# On the Interplay Between Misspecification and Sub-optimality Gap: From Linear Contextual Bandits to Linear MDPs

## Abstract

We study linear contextual bandits in the misspecified setting, where the expected reward function can be approximated by a linear function class up to a bounded misspecification level $\zeta > 0$. We propose an algorithm based on a novel data selection scheme, which only selects the contextual vectors with large uncertainty for online regression. We show that, when the misspecification level $\zeta$ is dominated by $\widetilde{\mathcal{O}}(\Delta/\sqrt{d})$ with $\Delta$ being the minimal sub-optimality gap and $d$ being the dimension of the contextual vectors, our algorithm enjoys the same gap-dependent regret bound $\widetilde{\mathcal{O}}(d^2/\Delta)$ as in the well-specified setting up to logarithmic factors. Together with a lower bound adapted from Du et al. (2019); Lattimore et al. (2020), our result suggests an interplay between misspecification level and the sub-optimality gap: (1) the linear contextual bandit model is efficiently learnable when $\zeta \leq \widetilde{\mathcal{O}}(\Delta/\sqrt{d})$; and (2) it is not efficiently learnable when $\zeta \geq \widetilde{\Omega}(\Delta/\sqrt{d})$. We also extend our algorithm to reinforcement learning with linear Markov decision processes (linear MDPs), and obtain a parallel result of gap-dependent regret. Experiments on both synthetic and real-world datasets corroborate our theoretical results.

## 1 Introduction

Linear contextual bandits (Li et al., 2010; Chu et al., 2011; Abbasi-Yadkori et al., 2011; Agrawal & Goyal, 2013) have been extensively studied when the reward function can be represented as a linear function of the contextual vectors. However, such a well-specified linear model assumption sometimes does not hold in practice. This motivates the study of misspecified linear models. In particular, we only assume that the reward function can be approximated by a linear function up to some worst-case error $\zeta$ called *misspecification level*. Existing algorithms for misspecified linear contextual bandits (Lattimore et al., 2020; Foster et al., 2020) can only achieve an $\widetilde{\mathcal{O}}(d\sqrt{K} + \zeta K\sqrt{d}\log K)$ regret bound, where $K$ is the total number of rounds and $d$ is the dimension of the contextual vector. Such a regret, however, suggests that the performance of these algorithms will degenerate to be linear in $K$ when $K$ is sufficiently large. The reason for this performance degeneration is because existing algorithms, such as OFUL (Abbasi-Yadkori et al., 2011) and linear Thompson sampling (Agrawal & Goyal, 2013), utilize all the collected data without selection. This makes these algorithms vulnerable to "outliers" caused by the misspecified model. Meanwhile, the aforementioned results do not consider the sub-optimality gap in the expected reward between the best arm and the second best arm. Intuitively speaking, if the sub-optimality gap is smaller than the misspecification level, there is no hope to obtain a sublinear regret. Therefore, it is sensible to take into account the sub-optimality gap in the misspecified setting, and pursue a gap-dependent regret bound.

The same misspecification issue also appears in reinforcement learning with linear function approximation, when a linear function cannot exactly represent the transition kernel or value function of the underlying MDP. In this case, Du et al. (2019) provided a negative result showing that if the misspecification level is larger than a certain threshold, any RL algorithm will suffer from an exponentially large sample complexity. This result was later revisited in the stochastic linear bandit setting by Lattimore et al. (2020), which shows that a large misspecification error will make the bandit model not efficiently learnable. However, these results cannot well explain the tremendous success of deep reinforcement learning on various tasks (Mnih et al., 2013; Schulman et al., 2015;

2017), where the deep neural networks are used as function approximators with misspecification error.

In this paper, we aim to understand the role of model misspecification in linear contextual bandits through the lens of sub-optimality gap. By proposing a new algorithm with data selection, we can achieve a constant regret bound for such a problem. We also extend our algorithm to the linear Markov decision processes (Jin et al., 2020) and obtain a regret bound of similar flavor. Our contributions are highlighted as follows:

- We propose a new algorithm called DS-OFUL (Data Selection OFUL). DS-OFUL only learns from the data with large uncertainty. We prove an $\widetilde{\mathcal{O}}(d^2\Delta^{-1})$ gap-dependent regret[1] bound when the misspecification level is small (i.e., $\zeta = \widetilde{\mathcal{O}}(\Delta/\sqrt{d})$) and the minimal sub-optimality gap $\Delta$ is known. Our regret bound improves upon the gap-dependent regret in the well-specified setting (Abbasi-Yadkori et al., 2011) by a logarithmic factor. To the best of our knowledge, this is the first constant gap-dependant regret bound for misspecified linear contextual bandits, even assuming a known minimal sub-optimality gap.

- We also prove a gap-dependent lower bound following the lower bound proof technique in Du et al. (2019); Lattimore et al. (2020). This together with the upper bound suggests an interplay between the misspecification level and the sub-optimality gap: the linear contextual bandit is efficiently learnable if $\zeta \leq \widetilde{\mathcal{O}}(\Delta/\sqrt{d})$ while it is not efficiently learnable if $\zeta \geq \widetilde{\Omega}(\Delta/\sqrt{d})$.

- We extend the same idea to the misspecified linear MDP, and propose an algorithm called DS-LSVI (Data-Selection LSVI). DS-LSVI enjoys a gap-dependent regret bound, which suggests a similar interplay between the misspecification level and sub-optimality gap in episodic MDPs to achieve a logarithmic regret bound $\widetilde{\mathcal{O}}(H^5 d^3 \Delta^{-1} \log(K))$

- Finally, we conduct experiments on the linear contextual bandit with both synthetic and real datasets, and demonstrate the superior performance of DS-OFUL algorithm. This corroborates our theoretical results.

**Notation.** Scalars and constants are denoted by lower and upper case letters, respectively. Vectors are denoted by lower case bold face letters $\mathbf{x}$, and matrices by upper case bold face letters $\mathbf{A}$. We denote by $[k]$ the set $\{1, 2, \cdots, k\}$ for positive integers $k$. For two non-negative sequence $\{a_n\}, \{b_n\}$, $a_n = \mathcal{O}(b_n)$ means that there exists a positive constant $C$ such that $a_n \leq Cb_n$, and we use $\widetilde{\mathcal{O}}(\cdot)$ to hide the log factor in $\mathcal{O}(\cdot)$ other than number of rounds $T$ or episode $K$; $a_n = \Omega(b_n)$ means that there exists a positive constant $C$ such that $a_n \geq Cb_n$, and we use $\widetilde{\Omega}(\cdot)$ to hide the log factor. For a vector $\mathbf{x} \in \mathbb{R}^d$ and a positive semi-definite matrix $\mathbf{A} \in \mathbb{R}^{d \times d}$, we define $\|\mathbf{x}\|_{\mathbf{A}}^2 = \mathbf{x}^\top \mathbf{A}\mathbf{x}$. For any set $\mathcal{C}$, we use $|\mathcal{C}|$ to denote its cardinality.

## 2 RELATED WORK

In this section, we review the related work for misspecified linear bandits and misspecified reinforcement learning. We defer more related work on the function approximation in bandits and RL to Appendix A.

**Misspecified Linear Bandits.** Ghosh et al. (2017) is probably the first work considering the misspecified linear bandits, which shows that the OFUL (Abbasi-Yadkori et al., 2011) algorithm cannot achieve a sublinear regret in the presence of misspecification. They, therefore, proposed a new algorithm with a hypothesis testing module for linearity to determine whether to use OFUL (Abbasi-Yadkori et al., 2011) or the multi-armed UCB algorithm. Their algorithm enjoys the same performance guarantee as OFUL in the well-specified setting and can avoid the linear regret under certain misspecification setting. Lattimore et al. (2020) proposed a phase-elimination algorithm for misspecified stochastic linear bandits, which achieves $\widetilde{\mathcal{O}}(\sqrt{dK} + \zeta K\sqrt{d})$ regret bound. For contextual linear bandits, both Lattimore et al. (2020) and Foster et al. (2020) proved a $\widetilde{\mathcal{O}}(d\sqrt{K} + \zeta K\sqrt{d})$ regret bound. Takemura et al. (2021); Vial et al. (2022) also provide a similar regret bound without the knowledge of the misspecification level. Van Roy & Dong (2019) proved a lower bound of sample complexity, which suggests when $\zeta\sqrt{d} \geq \sqrt{8 \log |\mathcal{D}|}$, any best arm identification algorithm will

---

[1] we use notation $\widetilde{\mathcal{O}}(\cdot)$ to hide the log factor other than number of rounds $T$

suffer a $\Omega(2^d)$ sample complexity, where $\mathcal{D}$ is the decision set. When the outcome is deterministic and does not contain noise, they provided an algorithm using $\widetilde{\mathcal{O}}(d)$ sample complexity to identify a $\Delta$-optimal arm when $\zeta \leq \Delta/\sqrt{d}$. Lattimore et al. (2020) also mentioned that if $\zeta\sqrt{d} \leq \Delta$, there exists a best arm identification algorithm can only use $\widetilde{\mathcal{O}}(d)$ arms to find a $\Delta$-optimal arm with the knowledge of $\zeta$. Note that although the exponential sample complexity lower bound for best arm identification can be translated into a regret lower bound in linear contextual bandits, the algorithms for best-arm identification and the corresponding upper bounds cannot be easily extended to linear contextual bandits. Besides these works on misspecification, He et al. (2022) studies the linear contextual bandits with adversarial corruptions, which can be considered as a similar setting of misspecification. They assume the summation of the approximation error over all $K$ rounds is bounded by the corruption level $C$. They proposed an algorithm achieving $\widetilde{\mathcal{O}}(d\sqrt{K} + dC)$ regret bound. However, their result in adversarial corrupted bandits cannot be directly translated to the misspecification setting since letting $C = K\zeta$ will lead to a $\mathcal{O}(d\sqrt{K} + dK\zeta)$ linear regret. Besides these series of work, Camilleri et al. (2021) also studies the robustness of kernel bandit algorithms on the misspecification.

**Misspecification in Reinforcement Learning** Du et al. (2019) showed that having a good representation is insufficient for efficient reinforcement learning unless the approximation error (i.e., misspecification level) by the representation is small enough. In particular, Du et al. (2019) suggested that a $\widetilde{\Omega}(\sqrt{H/d})$ misspecification will lead to $\Omega(2^H)$ sample complexity for RL to find the optimal policy, even with a generative model. On the other hand, a series of work (Jin et al., 2020; Zanette et al., 2020b; Foster & Rakhlin, 2020) provided $\widetilde{\mathcal{O}}(\sqrt{T} + \zeta T)$-type regret bound for RL in various settings, ignoring the dependence on the dimension of the feature representation $d$ and the planing horizon $H$. This suggests that the performance of RL will degenerate as the total number of interactions with the environment $T$ increases. Also, these results do not consider the minimal sub-optimality gap of the action-value function. Du et al. (2020) considered the agnostic Q-learning with misspecified linear function approximation. They proposed an algorithm with the access to a generative model and showed that if $\zeta \leq \widetilde{\mathcal{O}}(\Delta/\sqrt{d})$, one can find the optimal policy using $\mathcal{O}(d)$ trajectories. Together with the lower bound provided in Du et al. (2019), it suggests that $\zeta = \widetilde{O}(\Delta/\sqrt{d})$ is a sufficient and necessary condition to achieve a polynominal sample complexity given the access to the generative model.

## 3 PRELIMINARIES OF LINEAR CONTEXTUAL BANDITS

We consider a linear contextual bandit problem. In round $k \in [K]$, the agent receives a decision set $\mathcal{D}_k \subset \mathbb{R}^d$ and selects an arm $\mathbf{x}_k \in \mathcal{D}_k$ then observes the reward $r_k = r(\mathbf{x}_k) + \varepsilon_k$, where $r(\cdot) : \mathbb{R}^d \mapsto [0, 1]$ is a deterministic expected reward function and $\varepsilon_k$ is a zero-mean $R$-sub-Gaussian random noise. i.e., $\mathbb{E}[e^{\lambda \varepsilon_k} | \mathbf{x}_{1:k}, \varepsilon_{1:k-1}] \leq \exp(\lambda^2 R^2/2), \forall k \in [K], \lambda \in \mathbb{R}$.

In this work, we assume that all contextual vector $\mathbf{x} \in \mathcal{D}_k$ satisfies $\|\mathbf{x}\|_2 \leq L$ and the reward function $r(\cdot) : \mathbb{R}^d \to [0, 1]$ can be approximated by a linear function $r(\mathbf{x}) = \mathbf{x}^\top \boldsymbol{\theta}^* + \eta(\mathbf{x})$, where $\eta(\cdot) : \mathbb{R}^d \mapsto [-\zeta, \zeta]$ is the unknown misspecification error function. We further assume $\|\boldsymbol{\theta}^*\|_2 \leq B$ and for simplicity, we assume $B, L \geq 1$. We denote the optimal reward at round $k$ as $r_k^* = \max_{\mathbf{x} \in \mathcal{D}_k} r(\mathbf{x})$ and the optimal arm $\mathbf{x}_k^* = \operatorname{argmax}_{\mathbf{x} \in \mathcal{D}_k} r(\mathbf{x})$. Our goal is to minimize the regret defined by $\operatorname{Regret}(K) := \sum_{k=1}^K r_k^* - r(\mathbf{x}_k)$.

In this paper, we focus on the minimal sub-optimality gap condition.

**Definition 3.1** (Minimal sub-optimality gap). For each $\mathbf{x} \in \mathcal{D}_k$, the sub-optimality gap $\Delta_k(\mathbf{x})$ is defined by $\Delta_k(\mathbf{x}) := r_k^* - r(\mathbf{x})$ and the minimal sub-optimality gap $\Delta$ is defined by $\Delta := \min_{k \in [K], \mathbf{x} \in \mathcal{D}_k} \{\Delta_k(\mathbf{x}) : \Delta_k(\mathbf{x}) > 0\}$.

Then we further assume this minimal sub-optimality gap is strictly positive, i.e., $\Delta > 0$.

## 4 PROPOSED ALGORITHM

In this section, we propose our algorithm, DS-OFUL, in Algorithm 1. The algorithm runs for $K$ rounds. In each round, the algorithm first estimates the underlying parameter $\boldsymbol{\theta}^*$ by solving the

following ridge regression problem in Line 3

$$\boldsymbol{\theta}_k = \operatorname{argmin}_{\boldsymbol{\theta}} \sum_{i \in \mathcal{C}_{k-1}} \left( r_i - \mathbf{x}_i^\top \boldsymbol{\theta} \right)^2 + \lambda \|\boldsymbol{\theta}\|_2^2,$$

where $\mathcal{C}_{k-1}$ is the index set of the selected contextual vectors for regression and is initialized as an empty set at the beginning. After receiving the contextual vectors set $\mathcal{D}_k$, the algorithm selects an arm from the optimistic estimation powered by the Upper Confidence Bound (UCB) bonus in Line 4. In line 5, the algorithm adds the index of current round into $\mathcal{C}_k$ if the UCB bonus of the chosen arm $\mathbf{x}_k$, denoted by $\|\mathbf{x}_k\|_{\mathbf{U}_k^{-1}}$, is greater than the threshold $\Gamma$. Intuitively speaking, since the UCB bonus reflects the uncertainty of the model about the given arm $\mathbf{x}$, Line 5 discards the data that brings little uncertainty ($\|\mathbf{x}\|_{\mathbf{U}_k^{-1}}$) to the model. Finally we denote the total number of selected data in Line 5 by $|\mathcal{C}_K|$. We will declare the choices of the parameter $\Gamma, \beta$ and $\lambda$ in the next section.

---

**Algorithm 1** Data Selection OFUL (DS-OFUL)

---

**Input:** Threshold $\Gamma$, radius $\beta$ and regularizor $\lambda$
1: Initialize $\mathcal{C}_0 = \emptyset, \mathbf{U}_0 = \lambda \mathbf{I}, \boldsymbol{\theta}_0 = \mathbf{0}$
2: **for** $k = 1, \ldots, K$ **do**
3:     Set $\mathbf{U}_k = \lambda \mathbf{I} + \sum_{i \in \mathcal{C}_{k-1}} \mathbf{x}_i \mathbf{x}_i^\top, \boldsymbol{\theta}_k = \mathbf{U}_k^{-1} \sum_{i \in \mathcal{C}_{k-1}} r_i \mathbf{x}_i.$
4:     Receive decision set $\mathcal{D}_k$, select $\mathbf{x}_k = \operatorname{argmax}_{\mathbf{x} \in \mathcal{D}_k} \left\{ \mathbf{x}^\top \boldsymbol{\theta}_k + \beta \|\mathbf{x}\|_{\mathbf{U}_k^{-1}} \right\}$, receive reward $r_k$
5:     **if** $\|\mathbf{x}_k\|_{\mathbf{U}_k^{-1}} \geq \Gamma$ **then** $\mathcal{C}_k = \mathcal{C}_{k-1} \cup \{k\}$ **else** $\mathcal{C}_k = \mathcal{C}_{k-1}$
6: **end for**

---

## 5 REGRET ANALYSIS

In this section, we provide the regret upper bound of Algorithm 1 and the regret lower bound for learning the misspecified linear contextual bandit.

**Theorem 5.1** (Upper Bound)**.** For any $0 < \delta < 1$, let $\lambda = B^{-2}$ and $\Gamma = \Delta/(2\sqrt{d}\iota_1)$ where $\iota_1 = (24 + 18R)\log((72 + 54R)LB\sqrt{d}\Delta^{-1}) + \sqrt{8R^2 \log(1/\delta)}$. Set $\beta = 1 + 4\sqrt{d\iota_2} + R\sqrt{2d\iota_3}$ where $\iota_2 = \log(3LB\Gamma^{-1})$, $\iota_3 = \log((1 + 16L^2B^2\Gamma^{-2}\iota_2)/\delta)$. If the misspecification level is bounded by $2\sqrt{d}\zeta\iota_1 \leq \Delta$, then with probability at least $1 - \delta$, the cumulative regret of Algorithm 1 is bounded by

$$\operatorname{Regret}(K) \leq \frac{32\beta\sqrt{2d^3\iota_2 \log(1 + 16d\Gamma^{-2}\iota_2)}\iota_1}{\Delta}.$$

**Remark 5.2.** Since $\beta = \widetilde{\mathcal{O}}(\sqrt{d})$, Theorem 5.1 suggests an $\widetilde{\mathcal{O}}(d^2\Delta^{-1})$ constant regret bound independent of the total number of rounds $K$ when $\zeta \leq \widetilde{\mathcal{O}}(\Delta/\sqrt{d})$. This suggests an $\widetilde{\mathcal{O}}(d^2\Delta^{-1})$ constant regret bound if the misspecification level is reasonably small, which improves the logarithmic regret $\widetilde{\mathcal{O}}(d^2\Delta^{-1} \log(K))$ in Abbasi-Yadkori et al. (2011) to a constant regret[2]. Note that our constant regret bound relies on the knowledge of the minimal sub-optimality gap $\Delta$, while the OFUL algorithm in Abbasi-Yadkori et al. (2011) does not need prior knowledge about the minimal sub-optimality gap $\Delta$.

**Remark 5.3.** Our *high probability* constant regret bound does not violate the lower bound proved in Hao et al. (2020), which says that certain diversity condition on the contexts is necessary to achieve an *expected* constant regret bound (Papini et al., 2021). In contrast, we only provide a high-probability constant regret bound. When extending this high probability constant regret bound to expected regret bound, we have

$$\mathbb{E}[\operatorname{Regret}(K)] \leq \widetilde{\mathcal{O}}(d^2\Delta^{-1} \log(1/\delta))(1 - \delta) + \delta K,$$

which depends on $K$. To obtain a sub-linear expected regret, we can set $\delta = 1/K$ which yields a logarithmic regret $\widetilde{\mathcal{O}}(d^2\Delta^{-1} \log(K))$ and does not violate the lower bound in Hao et al. (2020).

---

[2]When we say constant regret, we ignore the $\log(1/\delta)$ factor in the regret as we choose $\delta$ to be a constant.

Furthermore, following the similar idea in Lattimore et al. (2020), we can prove a gap-dependent lower bound for misspecified stochastic linear bandits. Note that stochastic linear bandit can be seen as a special case of linear contextual bandits with a fixed decision set $\mathcal{D}_k = \mathcal{D}$ across all round $k \in [K]$. Similar result and proof can be found in Du et al. (2019) for episodic reinforcement learning.

**Theorem 5.4** (Lower Bound). Given the dimension $d$ and the number of arms $|\mathcal{D}|$, for any $\Delta \leq 1$ and $\zeta \geq 3\Delta\sqrt{8\log(|\mathcal{D}|)/(d-1)}$, there exists a set of stochastic linear bandit problems $\boldsymbol{\Theta}$ with minimal sub-optimality gap $\Delta$ and misspecification error level $\zeta$, such that for any algorithm that has a sublinear expected regret bound for all $\boldsymbol{\theta} \in \boldsymbol{\Theta}$, i.e., $\mathbb{E}[\text{Regret}_{\boldsymbol{\theta}}(K)] \leq CK^{\alpha}$ with $C > 0$ and $0 \leq \alpha < 1$, we have

- When $K \leq \mathcal{O}(|\mathcal{D}|)$, the expected regret is lower bounded by $\mathbb{E}_{\boldsymbol{\theta}\sim\text{Unif.}(\boldsymbol{\Theta})}[\text{Regret}_{\boldsymbol{\theta}}(K)] \geq K\Delta$.

- When $K \geq \Omega(|\mathcal{D}|)$, the expected regret is lower bounded by $\sup_{\boldsymbol{\theta}\in\boldsymbol{\Theta}} \mathbb{E}[\text{Regret}_{\boldsymbol{\theta}}(K)] \geq \widetilde{\Omega}(|\mathcal{D}|\log(K)\Delta^{-1})$.

**Remark 5.5.** Theorem 5.4 shows two regimes under the case $\zeta \geq \widetilde{\Omega}(\Delta/\sqrt{d})$. In the first regime $K \leq \mathcal{O}(|\mathcal{D}|)$ where the decision set is large (e.g., $|\mathcal{D}| = d^{100}$), any algorithm will suffer from a linear regret $\widetilde{\mathcal{O}}(\Delta K)$, which suggests that the regime cannot be efficiently learnable. In the second regime $K \geq \mathcal{O}(|\mathcal{D}|)$, Theorem 5.4 suggests a $\widetilde{\Omega}(|\mathcal{D}|\Delta^{-1}\log(K))$ regret lower bound, which is matched by the multi-armed bandit algorithm with an upper bound $\widetilde{\mathcal{O}}(|\mathcal{D}|\Delta^{-1}\log(K))$ (Lattimore & Szepesvári, 2020). Therefore, in this easier regime, linear function approximation cannot provide any performance improvement and one can simply adopt the multi-armed bandit algorithm to learn the bandit model.

**Remark 5.6.** Theorems 5.1 and 5.4 provide a holistic picture about the role of misspecification in linear contextual bandits. Here we focus on the more difficult regime $K \leq |\mathcal{D}|$. In the regime $K \leq |\mathcal{D}|$, when $\zeta \leq \widetilde{\mathcal{O}}(\Delta/\sqrt{d})$, Theorem 5.1 suggests that the bandit problem is efficiently learnable, and our algorithm DS-OFUL can achieve a constant regret, which improves upon the logarithmic regret bound in the well-specified setting (Abbasi-Yadkori et al., 2011). On the other hand, when $\zeta \geq \widetilde{\Omega}(\Delta/\sqrt{d})$, Theorem 5.4 provides a linear regret lower bound suggesting that the bandit model can not be efficiently learned.

## 6 PROOF SKETCH OF THEOREM 5.1

In this section, we give an overview of the main technical difficulty and our proof technique to derive Theorem 5.1. The detailed proof is deferred to Appendix C.

First, we aim at controlling the number of rounds in the index set $\mathcal{C}_K$. Since we only select the data with $\|\mathbf{x}_k\|_{\mathbf{U}_k^{-1}} \geq \Gamma$ for ridge regression, we can lower bound the summation of the selected UCB terms as $\sum_{k\in\mathcal{C}_K} \|\mathbf{x}_k\|_{\mathbf{U}_k^{-1}} \geq |\mathcal{C}_K|\Gamma$. On the other hand, noticing that $\mathbf{U}_k = \sum_{i\in\mathcal{C}_{k-1}} \mathbf{x}_k\mathbf{x}_k^{\top}$, we can upper bound the summation of UCB terms by using the elliptical potential lemma from Abbasi-Yadkori et al. (2011) as $\sum_{k\in\mathcal{C}_K} \|\mathbf{x}_k\|_{\mathbf{U}_k^{-1}} \leq \widetilde{\mathcal{O}}(\sqrt{d|\mathcal{C}_K|})$. Combining the upper bound and lower bound together we can bound the total number of the selected data $|\mathcal{C}_K|$ as $\Gamma|\mathcal{C}_K| \leq \widetilde{\mathcal{O}}(\sqrt{d|\mathcal{C}_K|})$, which suggests that $|\mathcal{C}_K| \leq \widetilde{\mathcal{O}}(d\Gamma^{-2})$ which is irreverent with the total number of rounds $K$.

Second, we control the fluctuations in the regression with misspecification error by $|\mathbf{x}^{\top}(\boldsymbol{\theta}_k - \boldsymbol{\theta}^*)| \leq \widetilde{\mathcal{O}}(R\sqrt{d}+\zeta\sqrt{|\mathcal{C}_K|})\|\mathbf{x}\|_{\mathbf{U}_k^{-1}}$. Compare this result with the original result $|\mathbf{x}^{\top}(\boldsymbol{\theta}_k - \boldsymbol{\theta}^*)| \leq \widetilde{\mathcal{O}}(R\sqrt{d}+\zeta\sqrt{dK})\|\mathbf{x}\|_{\mathbf{U}_k^{-1}}$ in Jin et al. (2020), our confidence radius $\widetilde{\mathcal{O}}(R\sqrt{d} + \zeta\sqrt{|\mathcal{C}_K|})$ does not grow with the total number of rounds $K$. In fact, directly use the result in Jin et al. (2020) and follow the proof outline in Abbasi-Yadkori et al. (2011) will lead to the following regret bound:

$$\text{Regret}(K) \leq \widetilde{\mathcal{O}}\left(R\sqrt{d} + \zeta\sqrt{dK}\right) \sum_{k=1}^{K} \|\mathbf{x}\|_{\mathbf{U}_k^{-1}} \leq \widetilde{\mathcal{O}}\left(Rd\sqrt{K} + \zeta dK\right),$$

which suggests a linear regret bound. As a comparison, with the help of data selection rule in our work, the regression set $\mathcal{C}_K$ is finite and we can avoid the linear regret when using all data into regression. In addition, our result provides a $\sqrt{d}$ tighter bound compared with Jin et al. (2020) by using the result provided in Zanette et al. (2020c)

Based on these two key observations, we overcome the linear regret bound by partitioning the total round $K$ into two different sets. The first set contains all non-selected round, i.e. $[K] \setminus \mathcal{C}_K$. In this situation, the uncertainty satisfies $\|\mathbf{x}_k\|_{\mathbf{U}_k^{-1}} < \Gamma$ and we can prove that when $\zeta \leq \widetilde{\mathcal{O}}(\Delta/\sqrt{d})$, the instantaneous regret for the un-selected round is bounded by

$$r_k^* - r(\mathbf{x}_k) \leq 2\zeta + 2\widetilde{\mathcal{O}}(R\sqrt{d} + \zeta\sqrt{d|\mathcal{C}_K|})\Gamma < \Delta,$$

which suggests that the non-selected data is optimal and incur no regret.

For the data in the finite set $\mathcal{C}_K$, we follows the gap-dependent regret bound in Abbasi-Yadkori et al. (2011) to show that $\sum_{k \in \mathcal{G}} r_k^* - r(\mathbf{x}_k) \leq \widetilde{\mathcal{O}}(d^2 \Delta^{-1}) \log(|\mathcal{C}_K|)$. As a result, by partition the set $[K]$ into two subsets $[K] \setminus \mathcal{C}_K, \mathcal{C}_K$, we get the claimed cumulative regret bound by

$$\text{Regret}(K) = \sum_{[K] \setminus \mathcal{C}_K} \text{Reg}(k) + \sum_{\mathcal{C}_K} \text{Reg}(k) \leq \widetilde{\mathcal{O}}\left(\frac{d^2 \log(|\mathcal{C}_K|)}{\Delta}\right) + 0,$$

where we denote $\text{Reg}(k)$ as the instantaneous regret in round $k$ (i.e. $r_k^* - r(\mathbf{x}_k)$).

# 7 MISSPECIFIED LINEAR MDPS

## 7.1 PRELIMINARIES OF LINEAR MDPS

We consider the episodic Markov Decision Process (MDP). Each episodic MDP is defined by a tuple $\mathcal{M}(\mathcal{S}, \mathcal{A}, H, \{r_h\}_{h=1}^H, \{P_h\}_{h=1}^H)$ where $\mathcal{S}$ is the state space, $\mathcal{A}$ is the action space, $H$ is the length of each episode and $r_h : \mathcal{S} \times \mathcal{A} \mapsto [0,1]$ is the reward function at stage $h$. $P_h$ is the transition kernel where $P_h(s'|s, a)$ denotes the transition probability from state $s$ to $s'$ with action $a$ at stage $h$. At the beginning of each episode, the agent determines a policy $\pi := \{\pi_h\}_{h=1}^H$ where $\pi_h : \mathcal{S} \mapsto \mathcal{A}$. Then from stage $h = 1$ to $h = H$, the agent repeatedly receives state $s_h$, takes the action $a_h = \pi_h(s_h)$, receives the reward $r_h(s_h, a_h)$ and the next state $s_{h+1}$. For any policy $\pi$, the value function and the $Q$-function at stage $h$ is defined by

$$V_h^\pi(s) = \mathbb{E}\left[\sum_{h'=h}^H r_{h'}(s_{h'}, \pi_{h'}(s_{h'})) \Big| s_h = s\right],$$
$$Q_h^\pi(s, a) = r_h(s, a) + \mathbb{E}\left[V_{h+1}^\pi(s_{h+1}) | s_h = s, a_h = a\right].$$

It's obvious that for all policy $\pi$, for all $h \in [H], s \in \mathcal{S}$ and $a \in \mathcal{A}$, the value function and the $Q$-function is bounded by $0 \leq V_h^\pi(s) \leq H, 0 \leq Q_h^\pi(s, a) \leq H$ since $r_h(s, a) \in [0, 1]$. We further define the optimal value function and the optimal $Q$-function as

$$V_h^*(s) = \max_\pi V_h^\pi(s), \quad Q_h^*(s, a) = \max_\pi Q_h^\pi(s, a).$$

For simplicity, we denote $[P_h V](s, a) = \mathbb{E}_{s' \sim P_h(\cdot|s,a)}[V(s')]$ and we have the Bellman equation along with the Bellman optimality equation as

$$Q_h^\pi(s, a) = r_h(s, a) + [P_h V_{h+1}^\pi](s, a), \quad Q_h^*(s, a) = r_h(s, a) + [P_h V_{h+1}^*](s, a). \tag{7.1}$$

We consider the $\zeta$-approximate linear MDP setting (Jin et al., 2020) in this work to study the impact of misspecification on function approximations in reinforcement learning.

**Definition 7.1** ($\zeta$-approximate linear MDP, Jin et al. 2020). For any $\zeta \leq 1$, we say that MDP $\mathcal{M}(\mathcal{S}, \mathcal{A}, H, \{r_h\}, \{P_h\})$ is a $\zeta$-approximate linear MDP with feature map $\phi : \mathcal{S} \times \mathcal{A} \mapsto \mathbb{R}^d$, if for any $h \in [H]$, there exists $d$ unknown (signed) measures $\boldsymbol{\mu}_h = \left(\mu_h^{(1)}, \cdots, \mu_h^{(d)}\right)$ over $\mathcal{S}$ and an unknown vector $\boldsymbol{\theta}_h^* \in \mathbb{R}^d$ such that for any $(s, a) \in \mathcal{S} \times \mathcal{A}$,

$$\|P_h(\cdot|s, a) - \langle \phi(s, a), \boldsymbol{\mu}_h(\cdot) \rangle\|_{\text{TV}} \leq \zeta, \quad |r_h(s, a) - \langle \phi(s, a), \boldsymbol{\theta}_h^* \rangle| \leq \zeta,$$

w.l.o.g. we assume $\|\phi(s, a)\|_2 \leq 1$ for all $(s, a) \in \mathcal{S} \times \mathcal{A}$ and $\max\{\|\boldsymbol{\mu}_h(\mathcal{S})\|_2, \|\boldsymbol{\theta}_h^*\|_2\} \leq \sqrt{d}$ for all $h \in [H]$.

Under Definition 7.1, it is easy to show that the $Q$-function under a certain policy $\pi$ is close to a linear function of the feature map $\phi$.

**Lemma 7.2** (Lemma C.1, Lemma C.2, Jin et al. 2020). *For a $\zeta$-approximate linear MDP, for any policy $\pi$, there exists a corresponding $\{\mathbf{w}_h^\pi\}_h$ such that for any $(s, a, h) \in \mathcal{S} \times \mathcal{A} \times [H]$:*

$$|Q_h^\pi(s,a) - \langle \phi(s,a), \mathbf{w}_h^\pi \rangle| \leq 2H\zeta, \|\mathbf{w}_h^\pi\|_2 \leq 2H\sqrt{d}.$$

We are concerning about minimizing the cumulative regret defined by $\text{Regret}(K) = \sum_{k=1}^K V_1^*(s_1^k) - V_1^{\pi^k}(s_1^k)$, where $\pi^k$ is the policy used in the $k$-th episode. Similar to linear contextual bandits, we introduce the minimal sub-optimality gap $\Delta$ originally defined in He et al. (2021a)

**Definition 7.3** (Minimal sub-optimality gap, He et al. 2021a). *For each $(s, a, h) \in \mathcal{S} \times \mathcal{A} \times [H]$, the sub-optimality gap $\Delta_h(s, a)$ is defined as $\Delta_h(s, a) := V_h^*(s) - Q_h^*(s, a)$ and the minimal sub-optimality gap is defined as $\Delta = \min_{h,s,a}\{\Delta_h(s, a) : \Delta_h(s, a) > 0\}$.*

We further assume this minimal sub-optimality gap is strictly positive, i.e., $\Delta > 0$.

## 7.2 Proposed Algorithm

We propose our algorithm, DS-LSVI, for misspecified linear MDP in Algorithm 2. It applies the idea of DS-OFUL to the LSVI-UCB (Jin et al., 2020). For simplicity, we denote $\phi_h^k = \phi(s_h^k, a_h^k), r_h^k = r_h(s_h^k, a_h^k)$ for short when there is no confusion. The algorithm runs for $K$ episodes. In the $k$-th episode, the algorithm estimates the optimal $Q$-function using a linear function as indicated by Lemma 7.2. In detail, at each stage $h$, after acquiring the estimated value function $V_{h+1}^k(\cdot)$ at state $h+1$, in Line 4, the algorithm solves the following ridge regression problem

$$\mathbf{w}_h^k = \text{argmin}_{\mathbf{w}} \|\mathbf{w}\|_2^2 + \sum_{i \in \mathcal{C}_{k-1}} \left( \langle \phi_h^i, \mathbf{w} \rangle - r_h^i - V_{h+1}^k(s_{h+1}^i) \right)^2,$$

where $\mathcal{C}_{k-1}$ contains the indices of episodes selected for regression and $r_h^i + V_{h+1}^k(s_{h+1}^i)$ is the estimated $Q$-function by Bellman optimality equation (7.1). Then the algorithm takes the greedy policy based on the estimated $Q$-function and receives the full episode. In Line 12, the algorithm adds the episode $k$ into the regression index set $\mathcal{C}_k$ if the data on $k$-th episode provides more uncertainty (i.e. $\|\phi\|_{\mathbf{U}^{-1}} \geq \Gamma$) at any stage $h \in [H]$. The intuition behind this selection is the same as Line 5 in Algorithm 1 when dealing with linear bandits: when one episode provide a data sample with large uncertainty at *any* stage, we add it to the regression. Otherwise, the episode will be ignored if the whole episode provide few uncertainty.

---

**Algorithm 2** Data Selection LSVI (DS-LSVI)

---

**Input:** Threshold $\Gamma$, radius $\beta$
1: Initialize $\mathcal{C}_0 = \emptyset, \mathbf{U}_h^0 = \mathbf{I}, \mathbf{w}_h^0 = \mathbf{0}$ for all $h \in [H], V_{H+1}^k(s) = Q_{H+1}^k(s, a) = 0$ for all $(s, a)$
2: **for** episodes $k = 1, \ldots, K$ **do**
3:     **for** stage $h = H, \ldots, 1$ **do**
4:         $\mathbf{U}_h^k = \mathbf{I} + \sum_{i \in \mathcal{C}_{k-1}} \phi_h^i(\phi_h^i)^\top$   $\mathbf{w}_h^k = (\mathbf{U}_h^k)^{-1} \sum_{i \in \mathcal{C}_{k-1}} \phi_h^i (r_h^i + V_{h+1}^k(s_{h+1}^i))$
5:         $Q_h^k(\cdot, \cdot) = \langle \phi(\cdot, \cdot), \mathbf{w}_h^k \rangle + \beta \|\phi(\cdot, \cdot)\|_{(\mathbf{U}_h^k)^{-1}}$
6:         $V_h^k(\cdot) = \min\{\max_a\{Q_h^k(\cdot, \cdot)\}, H\}, \pi_h^k(\cdot) = \text{argmax}_a\{Q_h^k(\cdot, a)\}$
7:     **end for**
8:     Receive initial state $s_1^k$
9:     **for** stage $h = 1, \ldots, H$ **do**
10:        Take action $a_h^k = \pi_h^k(s_h^k)$ and receive reward $r_h^k$ and next state $s_{h+1}^k$
11:     **end for**
12:     $\mathcal{C}_k = \mathcal{C}_{k-1} \cup \{k\}$ **if** $\exists h \in [H], \|\phi_h^k\|_{(\mathbf{U}_h^k)^{-1}} \geq \Gamma$ **else** $= \mathcal{C}_{k-1}$
13: **end for**

---

## 7.3 Regret Analysis

We provide the regret upper bound of Algorithm 2 for the $\zeta$-approximate linear MDP. The proof is deferred to Appendix F.

**Theorem 7.4** (Upper Bound). *Let $\Gamma = \widetilde{\Theta}(\Delta d^{-1} H^{-2}), \beta = \widetilde{\mathcal{O}}(Hd)$, with probability at least $1 - \delta$, if $\zeta = \widetilde{\mathcal{O}}(\Delta d^{-0.5} H^{-2.5})$, the cumulative regret of Algorithm 2 is bounded by $\text{Regret}(K) \leq \widetilde{\mathcal{O}}(H^5 d^3 \Delta^{-1} \log(K))$.*

**Remark 7.5.** Theorem 7.4 suggests that if the misspecification level $\zeta$ is upper bounded by $\mathcal{O}(\Delta d^{-0.5} H^{-2.5})$, we can achieve the same logarithmic regret bound $\widetilde{\mathcal{O}}(H^5 d^3 \Delta^{-1})$ as the well-specified setting (He et al., 2021a). This result indicates that a reasonably small misspecification will not deteriorate the performance. Our result improves the original regret bound $\widetilde{\mathcal{O}}(\sqrt{d^3 H^4 K} + \zeta d H^2 K)$ for the misspecified linear MDP provided in Jin et al. (2020).

**Remark 7.6.** Compared with the linear contextual bandit results, our result in the linear MDP setting also suggests the same relationship among $\zeta$, $\Delta$ and dimension $d$, ignoring the horizon factor $H$. Du et al. (2019) showed that when $\zeta \geq \widetilde{\Omega}(\Delta/\sqrt{d})$, any reinforcement learning algorithm suffers an $O(2^H)$ sample complexity. In addition, Du et al. (2020) provided algorithm for agnostic Q-learning which takes $\widetilde{\mathcal{O}}(d)$ trajectories to find the optimal policy when $\zeta < \Delta/\sqrt{d}$. However, their algorithm relies on a generative model which takes multiple actions $a$ for the same state $s$ at the same time. In contrast, Theorem 7.4 suggests that an $\widetilde{\mathcal{O}}(\Delta d^{-0.5} H^{-2.5})$ misspecification level can lead to a logarithmic regret without accessing the generative model.

## 8 EXPERIMENTS

To verify the performance improvement by data selection using the UCB bonus in Algorithm 1, we conduct experiments for bandit tasks on both synthetic and real-world datasets, which we will describe in detail below. We also carry out experiments for linear MDPs on a synthetic dataset in Appendix B.4.

Table 1: Averaged cumulative regret and elapsed time (E.T.) of DS-OFUL over 32 runs.

| $\Gamma$ | Regret (mean±std.) | E.T.(sec) |
|---|---|---|
| 0 [2] | $305.15 \pm 43.98$ | 10.26 |
| 0.02 | $332.28 \pm 76.17$ | 7.75 |
| 0.05 | $256.265 \pm 61.28$ | 5.96 |
| **0.08** | $\mathbf{184.75 \pm 61.91}$ | 5.21 |
| 0.20 | $374.63 \pm 277.51$ | **4.56** |
| LSW | $348.73 \pm 69.28$ | 4046 |
| RLB | $415.52 \pm 65.37$ | 8.66 |

[2] When $\Gamma = 0$, our algorithm degrades to OFUL
LSW: Using Eq. (6) in Lattimore et al. (2020)
RLB: Robust Linear Bandit (Ghosh et al., 2017)

### 8.1 SYNTHETIC DATASET

The synthetic dataset is composed as follows: we set $d = 16$ and generate parameter $\boldsymbol{\theta}^* \sim \mathcal{N}(\mathbf{0}, \mathbf{I}_d)$ and contextual vectors $\{\mathbf{x}_i\}_{i=1}^N \sim \mathcal{N}(\mathbf{0}, \mathbf{I}_d)$ where $N = 100$. The generated parameter and vectors are later normalized to be $\|\boldsymbol{\theta}^*\|_2 = \|\mathbf{x}_i\|_2 = 1$. The reward function is calculated by $r_i = \langle \boldsymbol{\theta}^*, \mathbf{x}_i \rangle + \eta_i$ where $\eta_i \sim \text{Unif}\{-\zeta, \zeta\}$. The contextual vectors and reward function is fixed after generated. The random noise on the receiving rewards $\varepsilon_t$ are sampled from the standard normal distribution.

We set the misspecification level $\zeta = 0.02$ and verified that the sub-optimality gap over the $N$ contextual vectors $\Delta \approx 0.18$. We do a grid search for $\beta = \{1, 3, 10\}$, $\lambda = \{1, 3, 10\}$ and report the cumulative regret of Algorithm 1 with different parameter $\Gamma = \{0, 0.02, 0.05, 0.08, 0.2\}$ over 32 independent trials with total rounds $K = 2000$. It's obvious that when $\Gamma = 0$, our algorithm degrades to the standard OFUL algorithm (Abbasi-Yadkori et al., 2011) which uses data from all rounds into regression.

The result is shown in Figure 1(b) and the average cumulative regret on the last round is reported in Table 1 with its variance over 32 trials. We can see that by setting $\Gamma \approx \Delta/\sqrt{d} \approx 0.18/\sqrt{16} \approx 0.05$, Algorithm 1 can achieve less cumulative regret compared with OFUL ($\Gamma = 0$). The algorithm with a proper choice of $\Gamma$ also convergences to zero instantaneous regret faster than OFUL. It is also evident that a slightly larger $\Gamma = 0.08$ will not affect the performance but a too large $\Gamma = 0.20 \geq \Delta$ will cause the algorithm to fail to learn the contextual vectors and induce a linear regret. Also, our algorithm shows that using a larger $\Gamma$ can significantly boost the speed of the algorithm by reducing the number of regressions needed in the algorithm.

We also compare with the algorithm (LSW) in Equation (6) of Lattimore et al. (2020) and the RLB in Ghosh et al. (2017) in Figure 1(b) and Table 1. For Lattimore et al. (2020), the estimated reward is updated by $r(\mathbf{x}) = \mathbf{x}^\top \boldsymbol{\theta}_k + \beta \|\mathbf{x}\|_{\mathbf{U}_k^{-1}} + \varepsilon \sum_{s=1}^k |\mathbf{x}^\top \mathbf{U}_k^{-1} \mathbf{x}_s^{-1}|$. However, since the term $\varepsilon \sum_{s=1}^k |\mathbf{x}^\top \mathbf{U}_k^{-1} \mathbf{x}_s^{-1}|$ is hard to be updated incrementally w.r.t. $k$, this algorithm is less efficient than OFUL Abbasi-Yadkori et al. (2011) as well as our algorithm. For the RLB algorithm in Ghosh et al. (2017), we did the hypothesis test for $k = 10$ rounds and then decided whether to use OFUL

or multi-armed UCB. The results show that both LSW and RLB achieve a worse regret than OFUL since in our setting $\zeta$ is relatively small.

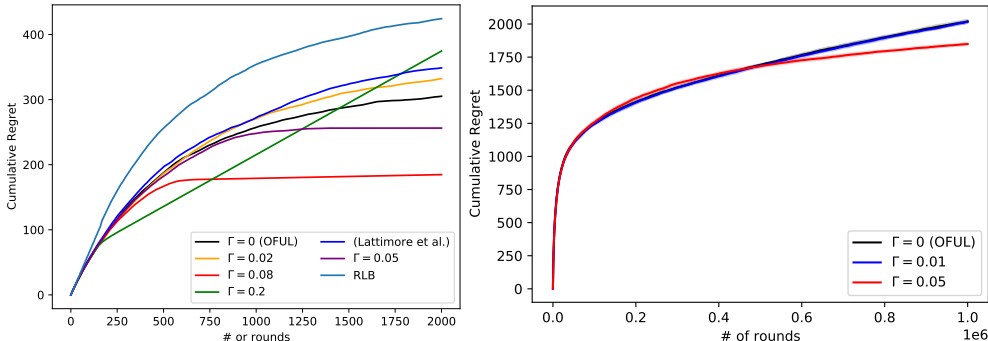

(a) Cumulative regret comparison of DS-OFUL (with difference choices of $\Gamma$), Lattimore et al. (2020) and RLB over 2000 rounds. Results are averaged over 32 replicates.

(b) Cumulative regret of DS-OFUL on the Asirra dataset over 1M rounds with different $\Gamma$ under misspecification level $\zeta = 0.01$. Results are averaged over 8 runs with standard errors shown as shaded areas.

Figure 1: Experiment results on (a): synthetic dataset, and (b): real-world dataset.

## 8.2 REAL-WORLD DATASET

To demonstrate that the proposed algorithm can be easily applied to modern machine learning tasks, we carried out experiments on the Asirra dataset (Elson et al., 2007). The task of agent is to distinguish the image of cats from the image of dogs. At each round $k$, the agent receives the feature vector $\phi_{1,k} \in \mathbb{R}^{512}$ of a cat image and another feature vector $\phi_{2,k} \in \mathbb{R}^{512}$ of a dog image. Both feature vectors are generated using ResNet-18 (He et al., 2016) pretrained on ImageNet (Deng et al., 2009). We normalize $\|\phi_{1,k}\|_2 = \|\phi_{2,k}\|_2 = 1$. The agent is required to select the cat from these two vectors. It receives reward $r_t = 1$ if it selects the correct feature vector, and receives $r_t = 0$ otherwise. It is trivial that the sub-optimality gap of this task is $\Delta = 1$. To better demonstrate the influence of misspecification on the performance of the algorithm, we only select the data with $|\phi_i^\top \theta^* - r_i| \leq \zeta$ with $r_i = 1$ if it is a cat and $r_i = 0$ otherwise. $\theta^*$ is a pretrained parameter on the whole dataset using linear regression $\theta^* = \operatorname{argmin}_{\theta} \sum_{i=1}^{N} (\phi_i^\top \theta - r_i)^2$, which the agent does not know.

For hyper-parameter tuning, we select $\beta = \{1, 0.3, 0.1\}$ and $\lambda = \{1, 3, 10\}$ by doing a grid search and repeat the experiments for 8 times over 1M rounds for each parameter configuration. As shown in Figure 1(a), when $\zeta = 0.01$, though the OFUL algorithm (setting $\Gamma = 0$) will have a better performance at the very beginning, setting $\Gamma = 0.05 \approx \Delta/\sqrt{d} = 1/\sqrt{512}$ will eventually improve the performance of the algorithm. As a sensitivity analysis, we also set $\zeta = \{0.5, 0.1, 0.05\}$ to test the impact of misspecification on the performance of algorithm choices of $\Gamma$. More experiment configurations and results are deferred to Appendix B.

## 9 CONCLUSION AND FUTURE WORK

We study the misspecified linear contextual bandit from a gap-dependent perceptive. We propose an algorithm and show that if the misspecification level $\zeta \leq \widetilde{\mathcal{O}}(\Delta/\sqrt{d})$, the proposed algorithm can achieve the same gap-dependent regret bound as in the well-specified case. Along with Lattimore et al. (2020); Du et al. (2019), we provide a complete picture on the interplay between misspecification and sub-optimality gap, in which $\Delta/\sqrt{d}$ plays an important role on the phase transition of $\zeta$ to decide if the bandit model can be efficiently learned. The algorithm and analysis have been extended to linear Markov decision processes and verified via experiments as well.

The promising result suggests a few interesting directions for future research. For example, it remains unclear if we can get rid of the prior knowledge of the minimum sub-optimality gap to achieve similar regret guarantees. It would also be interesting to incorporate the Lipschitz continuity or smoothness properties of the reward function to derive fine-grained results.

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

## A  ADDITIONAL RELATED WORK

**Linear Contextual Bandits.** There is a large body of literature on linear contextual bandits. For example, Auer (2002); Chu et al. (2011); Agrawal & Goyal (2013) studied linear contextual bandits when the number of arms is finite. Abbasi-Yadkori et al. (2011) proposed an algorithm called OFUL to deal with the infinite arm set. All these works come with an $\widetilde{\mathcal{O}}(\sqrt{K})$ problem-independent regret bound, and an $\mathcal{O}(d^2 \Delta^{-1} \log(K))$ gap-dependent regret bound is also given by Abbasi-Yadkori et al. (2011).

**RL with Linear Function Approximation.** To tackle RL tasks in large state space, a line of work on RL with linear function approximation has emerged in the past years. For example, linear MDPs (Yang & Wang, 2019; Jin et al., 2020) is probably one of the most widely studied models where both the transition kernel and the reward function are linear functions of a known feature mapping. Typical algorithms in this setting include LSVI-UCB (Jin et al., 2020) and randomized LSVI (Zanette et al., 2020a), both of which can achieve a sublinear regret. Besides, linear mixture/kernel MDPs (Modi et al., 2020; Jia et al., 2020; Ayoub et al., 2020; Zhou et al., 2021) has emerged as a popular model for model-based RL with linear function approximation, in which the transition kernel is defined as a mixture of feature mappings defined on the triplet of state, action, and next state. In this setting, nearly minimax optimal regret has been attained for both finite-horizon episodic MDPs and infinite-horizon discounted MDPs (Zhou et al., 2021). The aforementioned works are focused on the problem-independent regret bound, while He et al. (2021a) provided a gap-dependent regret bound for both linear MDPs (i.e., $\widetilde{\mathcal{O}}(d^3 H^5 \Delta^{-1})$) and linear mixture MDPs (i.e., $\widetilde{\mathcal{O}}(d^2 H^5 \Delta^{-1})$).

## B  EXPERIMENT DETAILS AND ADDITIONAL RESULTS

### B.1  EXPERIMENT CONFIGURATION

The experiment on synthetic dataset is conducted on Google Colab with a 2-core Intel® Xeon® CPU @ 2.20GHz. The experiment on the real-world Asirra dataset (Elson et al., 2007) is conducted on an AWS p2-xlarge instance.

### B.2  DATA PREPROCESSING FOR THE ASIRRA DATASET

To demonstrate how our algorithm can deal with different levels of misspecification, we do data preprocessing before feeding the data into the agent. As described in Section 8.2, the remaining data with expected misspecification level $\zeta$ are shown in Table 2. It can be verified that even with the smallest misspecification level, there are still more than 10% of the data is selected.

Table 2: The number of remaining data samples after data processing with expected misspecification level

| $\zeta$ | # of cats | # of dogs |
|---|---|---|
| $\infty$ (without preprocessing) | 12500 | 12500 |
| 0.5 (linear separable) | 10316 | 10511 |
| 0.1 | 3182 | 3248 |
| 0.05 | 2408 | 2442 |
| 0.01 | 1886 | 1905 |

### B.3  ADDITIONAL RESULT ON THE ASIRRA DATASET

As a sensitivity analysis, we change the misspecification level in the preprocessing part in the Asirra dataset. The result is shown in Figure 2. This result suggests that when the misspecification is small enough, setting $\Gamma = \Delta/\sqrt{d}$ can deliver a reasonable result. It is aligned with the parameter setting in our theorem. Meanwhile, we found that when $\zeta = 0.5$, which means it is strictly larger than the threshold $\Delta/\sqrt{d}$, the algorithm cannot achieve a similar performance with of $\zeta < 0.1$, regardless of the setting of parameter $\Gamma$. This also verifies the theoretical understanding of how a large misspecification level will harm the performance of the algorithm.

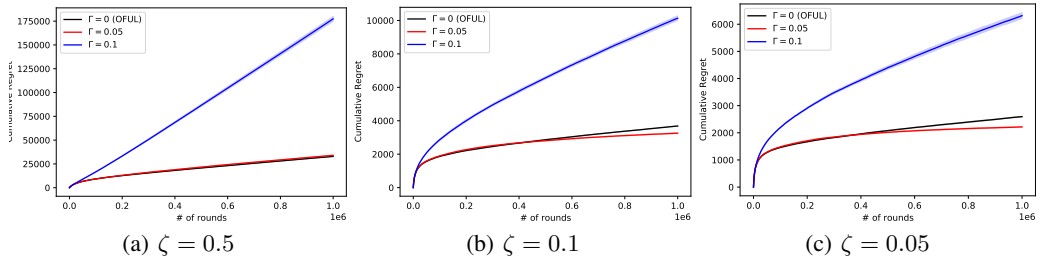

Figure 2: The performance of DS-OFUL under different misspecification level $\zeta$. Results are averaged over 8 runs with standard errors shown as shaded areas.

### B.4 ADDITIONAL EXPERIMENTS ON MISSPECIFIED LINEAR MDPs

To further verify the performance of our algorithm in misspecified linear MDPs, we generate a synthetic MDP as follows. We select $S = 6, A = 4, H = 2$ and generate $\boldsymbol{\phi}(s, a)$ and $\boldsymbol{\mu}(s') \sim$ Unif.$(\mathbf{0}_d, \mathbf{1}_d)$ where $d = 3$ respectively. Since $H = 2$, we define the first-stage transition kernel as

$$P_1(s'|s, a) = \langle \boldsymbol{\phi}(s, a), \boldsymbol{\mu}_h(s') \rangle + \eta(s')/S$$

with normalization $\boldsymbol{\phi}(s, a) = \boldsymbol{\phi}(s, a)/\sum_{s \in [S]} \langle \boldsymbol{\phi}(s, a), \boldsymbol{\mu}_h(s') \rangle$. The misspecification error $\eta(s')$ satisfies that $|\eta(s')| \leq 2 \times 10^{-3}$ and $\sum_{s' \in [S]} \eta(s') = 0$. The reward is generated by $r(s, a) = \langle \boldsymbol{\phi}(s, a), \boldsymbol{\theta} \rangle + \eta(s, a)$ with $|\eta(s, a)| \leq 5 \times 10^{-3}$. It is easy to verify that the generated MDP is a $\zeta$-approximate linear MDP with $\zeta = 5 \times 10^{-3}$ according to Definition 7.1. Then with the true transition kernel and the reward function, we can calculate that the minimal sub-optimality gap for the generated MDP is $\Delta \approx 0.0101$.

According to our theory, we choose $\beta = \{1, 3, 10\}$ and report the cumulative regret with different choices of $\Gamma$. The results are shown in Table 3 and Figure 3. Three key observations can be revealed from the experiment result in Table 3. First, choosing $\Gamma = 0.01$ can achieve the best performance (lowest cumulative regret). Second, choosing $0.005 \leq \Gamma \leq 0.01$ can also lead to a comparable constant regret, according to Table 3, but smaller $\Gamma$ may not lead to zero instantaneous regret within 200K rounds. Third, setting $\Gamma > \Delta$ will lead to a linear regret.

In addition, our algorithm degrades to LSVI-UCB algorithm Jin et al. (2020) by setting $\Gamma = 0$. For misspecified linear MDPs, the algorithm studied in Theorem 3.2 of Jin et al. (2020) is still LSVI-UCB with a different choices of confidence radius $\beta$. In our experiments the $\beta$ is tuned as a hyper-parameter. Therefore, the experiment results suggest that our algorithm outperforms the LSVI-UCB algorithm in the misspecified setting.

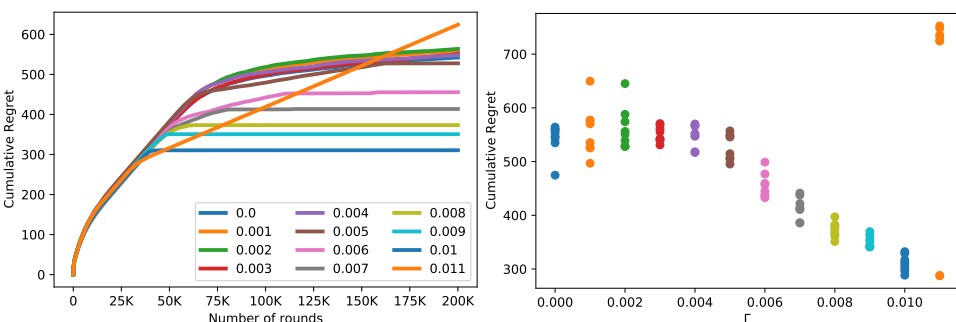

(a) Averaged cumulative regret of DS-LSVI over 8 replicates with different choices of $\Gamma$.

(b) Cumulative regret of DS-LSVI with different choices of $\Gamma$ after 200K rounds. Each experiment with the same $\Gamma$ is repeated for 8 times, and each point represents an individual experiment.

Figure 3: The performance of DS-LSVI with different choices of $\Gamma$. (a): averaged cumulative regret w.r.t number of rounds. (b): cumulative regret after 200K rounds with different choices of $\Gamma$.

Table 3: Averaged cumulative and instantaneous regret of DS-LSVI over 8 replicates for 200K rounds with different choice of $\Gamma$ (see the footnote for the specific choice of $\Gamma$). "Last 1K rounds cumulative regret" means the total regret incurred in the very last 1K rounds, in order to verify the final performance of the algorithm. "Total 200K rounds cumulative regret" means the total regret incurred during the entire 200K rounds.

| $\Gamma$ | Last 1K rounds cumulative regret (mean±std.) | Total 200K round cumulative regret (mean±std.) |
|---|---|---|
| $0.0^1$ | $0.133 \pm 0.291$ | $541.9 \pm 27.0$ |
| 0.001 | $0.118 \pm 0.215$ | $557.1 \pm 44.2$ |
| 0.002 | $0.295 \pm 0.480$ | $563.7 \pm 36.5$ |
| 0.003 | $0.523 \pm 0.323$ | $551.2 \pm 13.6$ |
| 0.004 | $0.185 \pm 0.133$ | $548.3 \pm 19.5$ |
| $0.005^2$ | $\mathbf{0.00 \pm 0.00}$ | $527.7 \pm 23.0$ |
| 0.006 | $\mathbf{0.00 \pm 0.00}$ | $455.6 \pm 21.4$ |
| 0.007 | $\mathbf{0.00 \pm 0.00}$ | $413.6 \pm 19.2$ |
| 0.008 | $\mathbf{0.00 \pm 0.00}$ | $373.4 \pm 13.3$ |
| 0.009 | $\mathbf{0.00 \pm 0.00}$ | $350.7 \pm 10.3$ |
| 0.01 | $\mathbf{0.00 \pm 0.00}$ | $\mathbf{310.6 \pm 14.3}$ |
| 0.011 | $2.053 \pm 1.185$ | $624.2 \pm 194.5$ |

[1] When $\Gamma = 0$, our algorithm degrades to LSVI-UCB
[2] $\Gamma = 0.005 \approx \Delta/\sqrt{d} = 0.0101/\sqrt{3}$ in this setting

## C    DETAILED PROOF OF THEOREM 5.1

In this section, we provide the detailed proof for Theorem 5.1. First, we present a technical lemma to bound the total number of data used in the online linear regression in Algorithm 1.

**Lemma C.1.** Given $0 < \Gamma \leq 1$, set $\lambda = B^{-2}$. For any $k \in [K]$, $|\mathcal{C}_k| \leq 16d\Gamma^{-2}\log(3LB\Gamma^{-1})$.

Lemma C.1 suggests that up to $\widetilde{O}(d\Gamma^{-2})$ contextual vectors has a UCB bonus greater than $\Gamma$. A similar result is also provided in He et al. (2021b), suggesting a $\widetilde{\mathcal{O}}(\Gamma^{-2})$ Uniform-PAC sample complexity. Lemma C.1 also suggests that the numbers of data added in regression set $\mathcal{C}$ is finite, thus the regression procedure is not affected critically by the noise and the misspecification error.

For a linear regression with up to $|\mathcal{C}_k|$ data, the next lemma is crucial in controlling the fluctuations with misspecification error.

**Lemma C.2.** Let $\lambda = B^{-2}$. For all $\delta > 0$, with probability at least $1 - \delta$, for all $\mathbf{x} \in \mathbb{R}^d, k \in [K]$, the estimation error is bounded by:

$$|\mathbf{x}^\top(\boldsymbol{\theta}_k - \boldsymbol{\theta}^*)| \leq \left(1 + R\sqrt{2d\iota} + \zeta\sqrt{|\mathcal{C}_k|}\right)\|\mathbf{x}\|_{\mathbf{U}_k^{-1}},$$

where $\iota = \log((d + |\mathcal{C}_k|L^2B^2)/(d\delta))$ and $|\mathcal{C}_k|$ is the total number of data used in regression at $k$-th round.

Lemma C.2 provides a similar decomposition as the well-specified linear contextual bandits algorithms like OFUL (Abbasi-Yadkori et al., 2011). However, comparing the confidence radius here $\widetilde{\mathcal{O}}(R\sqrt{d} + \zeta\sqrt{|\mathcal{C}_{k-1}|})$ with the conventional radius in OFUL $\widetilde{\mathcal{O}}(R\sqrt{d})$, one can find that the misspecification term will affect the radius in an $\sqrt{|\mathcal{C}_K|}$ order. If we directly use all data to do the regression, the confidence radius will be in the order of $\widetilde{\mathcal{O}}(\sqrt{K})$ and therefore would lead to a $\mathcal{O}(K\sqrt{\log K})$ regret bound (see Lemma 11 in Abbasi-Yadkori et al. (2011)). This makes the regret bound trivial since it goes beyond the trivial regret upper bound $\mathcal{O}(K)$ when $K$ grows larger. In ours, however, the confidence radius is only $\sqrt{|\mathcal{C}_K|}$ where $|\mathcal{C}_K|$ is finite given Lemma C.1. As a result, our regret bound will not grow with $K$ as OFUL, and will have a more stable prediction.

When the misspecification level is well bounded by $\zeta = \widetilde{\mathcal{O}}(\Delta/\sqrt{d})$, the following corollary is a direct result of Lemmas C.2 by replacing the term $|\mathcal{C}_K|$ with its upper bound provided in Lemma C.1.

**Corollary C.3.** Suppose $2\sqrt{d}\zeta\iota_1 \leq \Delta$, let $\lambda = B^{-2}$ and $0 < \Gamma \leq 1$. Let $\beta = 1 + 2\Delta\Gamma^{-1}\sqrt{\iota_2}/\iota_1 + R\sqrt{2d\iota_3}$ where $\iota_2 = \log(3LB\Gamma^{-1})$, $\iota_3 = \log((1+16L^2B^2\Gamma^{-2}\iota_2)/\delta)$, then with probability at least

$1-\delta$, for all $\mathbf{x} \in \mathbb{R}^d, k \in [K]$, the estimation error at round $k \in [K]$ is bounded by: $|\mathbf{x}^\top(\boldsymbol{\theta}_k - \boldsymbol{\theta}^*)| \leq \beta \|\mathbf{x}\|_{\mathbf{U}_k^{-1}}$.

*Proof.* By Lemma C.1, replacing $|\mathcal{C}_K|$ with its upper bound yields

$$|\mathbf{x}^\top(\boldsymbol{\theta}_k - \boldsymbol{\theta}^*)| \leq (1 + 4\sqrt{d}\zeta\Gamma^{-1}\sqrt{\iota_2} + R\sqrt{2d\iota_3})\|\mathbf{x}\|_{\mathbf{U}_k^{-1}} \leq \beta\|\mathbf{x}\|_{\mathbf{U}_k^{-1}},$$

where the second inequality is due to the condition $2\sqrt{d}\zeta \leq \Delta/\iota_1$. $\qquad\square$

Next we introduce an auxiliary lemma controlling the instantaneous regret bound using the UCB bonus and the misspecification level.

**Lemma C.4.** Suppose Corollary C.3 holds, for all $k \in [K]$, the instantaneous regret at round $k$ is bounded by

$$\Delta_k(\mathbf{x}_k) = r_k^* - r(\mathbf{x}_k) \leq 2\zeta + 2\beta\|\mathbf{x}_k\|_{\mathbf{U}_k^{-1}}.$$

The next auxiliary lemma from He et al. (2021a) bounds the summation of a subset of the self-normalized vectors.

**Lemma C.5** (Lemma 6.6, He et al. 2021a)**.** For any subset $\mathcal{G} = \{c_1, \cdots, c_i\} \subseteq \mathcal{C}_K$, we have

$$\sum_{k \in \mathcal{G}} \|\mathbf{x}_k\|_{\mathbf{U}_k^{-1}}^2 \leq 2d\log(1 + |\mathcal{G}|L^2/\lambda).$$

The next two technical algebra lemma is used to control the dominated terms

**Lemma C.6.** Let $\iota_1 = (24+18R)\log((72+54R)LB\sqrt{d}\Delta^{-1}) + \sqrt{8R^2\log(1/\delta)}, \Gamma = \Delta/(2\sqrt{d}\iota_1)$, $\iota_2 = \log(3LB\Gamma^{-1}), \iota_3 = \log((1 + 16L^2B^2\Gamma^{-2}\iota_2)/\delta)$, we have $\iota_1 > 2 + 4\sqrt{\iota_2} + R\sqrt{2\iota_3}$.

Equipped with these lemmas, we can start the proof of Theorem 5.1.

*Proof of Theorem 5.1.* First, it worth mentioning that by setting $\Gamma = \Delta/(2\sqrt{d}\iota_1)$, the confidence radius $\beta$ becomes $1 + 4\sqrt{d\iota_2} + R\sqrt{2d\iota_3}$. Then our proof starts with assuming that Corollary C.3 holds with probability at least $1 - \delta$. We decompose the index set $[K]$ into two subsets. The first set is $[K] \setminus \mathcal{C}_K$ indicating the non-selected data, the second set is the selected set $\mathcal{C}_K$. We will bound the cumulative regret within two set separately.

First, for those non-selected data $k \notin \mathcal{C}_k$, i.e. $\|\mathbf{x}_k\|_{\mathbf{U}_k^{-1}} < \Gamma$, combining Lemma C.4 with Corollary C.3 yields

$$r_k^* - r(\mathbf{x}_k) < 2\zeta + 2\beta\Gamma = 2\zeta + \frac{\Delta}{\sqrt{d}\iota_1} + \frac{\sqrt{2\iota_3}R\Delta}{\iota_1} + \frac{4\Delta\sqrt{\iota_2}}{\iota_1}, \tag{C.1}$$

where $\iota_1, \iota_2, \iota_3$ are the same as Theorem 5.1, and the second equation is from $\Gamma = \Delta/(2\sqrt{d}\iota_1)$. When misspecification condition $2\sqrt{d}\zeta \leq \Delta/\iota_3$ holds, (C.1) suggests that

$$r_k^* - r(\mathbf{x}_k) < \frac{2\Delta}{\sqrt{d}\iota_1} + \frac{4\Delta\sqrt{\iota_2}}{\iota_1} + \frac{\sqrt{2\iota_3}R\Delta}{\iota_1}. \tag{C.2}$$

Lemma C.6 suggests that when $\iota_1 = (24 + 18R)\log((72 + 54R)LB\sqrt{d}\Delta^{-1}) + \sqrt{8R^2\log(1/\delta)}$ $\iota_1 > 2 + 4\sqrt{\iota_2} + R\sqrt{2\iota_3}$, (C.2) yields that the instantaneous regret $r_k^* - r(\mathbf{x}_k) < \Delta$ at round $k$. By Definition 3.1, the instantaneous regret is zero for all $k \notin \mathcal{C}_k$, indicating the non-selected data can incur zero instantaneous regret.

Then Lemma C.4 suggests that the instantaneous regret for those $k \in \mathcal{C}_K$ is bounded by

$$
\begin{aligned}
\sum_{k \in \mathcal{C}_K} r_k^* - r(\mathbf{x}_k) &\le \sum_{k \in \mathcal{C}_K} \left( 2\beta \|\phi_k\|_{\mathbf{U}_k^{-1}} + 2\zeta \right) \\
&\le 2\beta \sqrt{|\mathcal{C}_K|} \sqrt{\sum_{k \in \mathcal{C}_K} \|\phi_k\|_{\mathbf{U}_k^{-1}}^2} + 2|\mathcal{C}_K|\zeta \\
&\le 8\beta\Gamma^{-1}\sqrt{d\iota_2}\sqrt{2d \log(1 + 16d\Gamma^{-2}\iota_2)} + 32\zeta d\Gamma^{-2}\iota_2 \\
&\le 16\beta\sqrt{2d^3\iota_2 \log(1 + 16d\Gamma^{-2}\iota_2)}\iota_1/\Delta + 64\sqrt{d^3}\iota_1\iota_2/\Delta \\
&\le 32\beta\sqrt{2d^3\iota_2 \log(1 + 16d\Gamma^{-2}\iota_2)}\iota_1/\Delta,
\end{aligned}
\tag{C.3}
$$

where the second inequality follows the C.S. inequality, the third one yields from Lemma C.5 while the fourth utilize the fact that $\Gamma = \Delta/(2\sqrt{d}\iota_1)$ and $\zeta \le \Delta/(2\sqrt{d}\iota_1)$. The last one is due to the fact that the second term in the forth inequality is dominated by the first one.

To warp up, the cumulative regret can be decomposed by

$$
\text{Regret}(K) = \sum_{k \notin \mathcal{C}_K} (r_k^* - r(\mathbf{x}_k)) + \sum_{k \in \mathcal{C}_K} (r_k^* - r(\mathbf{x}_k)) \le 0 + \frac{32\beta\sqrt{2d^3\iota_2 \log(1 + 16d\Gamma^{-2}\iota_2)}\iota_1}{\Delta},
$$

where the first two zeros are given by the fact that for $k \notin \mathcal{C}_K$, we have $r_k^* - r(\mathbf{x}_k) = 0$. the regret bound for $k \in \mathcal{G}$ is given by (C.3). $\qquad\square$

# D   PROOF OF TECHNICAL LEMMAS IN APPENDIX C

## D.1   PROOF OF LEMMA C.1

To prove this lemma, we introduce the well known elliptical potential lemma by Abbasi-Yadkori et al. (2011)

**Lemma D.1** (Lemma 11, Abbasi-Yadkori et al. 2011). Let $\{\phi_i\}_{i=1}^I$ be a sequence in $\mathbb{R}^d$, define $\mathbf{U}_i = \lambda \mathbf{I} + \sum_{j=1}^i \phi_j \phi_j^\top$, then

$$
\sum_{i=1}^I \min\left\{ 1, \|\phi_i\|_{\mathbf{U}_{i-1}^{-1}}^2 \right\} \le 2d \log\left( \frac{\lambda d + IL^2}{\lambda d} \right).
$$

Then the following auxiliary lemma and its corollary are useful

**Lemma D.2** (Lemma A.2, Shalev-Shwartz & Ben-David 2014). Let $a \ge 1$ and $b > 0$. Then $x \ge 4a \log(2a) + 2b$ yields $x \ge a \log(x) + b$.

Lemma D.2 can easily indicate the following lemma.

**Lemma D.3.** Let $a \ge 1$. Then $x \ge 4 \log(2a) + a^{-1}$ yields $x \ge \log(1 + ax)$.

*Proof.* Let $y = 1 + ax, x = (y - 1)/a$. Then $x \ge 4 \log(2a) + a^{-1}$ is equivalent with $y \ge 4a \log(2a) + 2$. By Lemma D.2, this implies $y \ge a \log(y) + 1$ which is exactly $x \ge \log(1 + ax)$. $\quad\square$

Equipped with these technical lemmas, we can start our proof.

*Proof of Lemma C.1.* Since the cardinality of set $\mathcal{C}_k$ is monotonically increasing w.r.t. $k$, we fix $k$ to be $K$ in the proof and only provide the bound of $\mathcal{C}_K$. For all selected data $k \in \mathcal{C}_K$, we have $\|\phi_k\|_{\mathbf{U}_k^{-1}} \ge \Gamma$. Therefore when $\Gamma \le 1$, the summation of the UCB bonus over data $k \in \mathcal{C}_K$ is lower bounded by

$$
\sum_{k \in \mathcal{C}_K} \min\left\{ 1, \|\phi_k\|_{\mathbf{U}_k^{-1}}^2 \right\} \ge |\mathcal{C}_K| \min\{1, \Gamma^2\} = |\mathcal{C}_K|\Gamma^2.
\tag{D.1}
$$

On the other hand, Lemma D.1 implies

$$\sum_{k \in \mathcal{C}_K} \min \left\{ 1, \|\phi_k\|^2_{\mathbf{U}_k^{-1}} \right\} \leq 2d \log \left( \frac{\lambda d + |\mathcal{C}_K| L^2}{\lambda d} \right). \tag{D.2}$$

Combining (D.2) and (D.1), the total number of the selected samples $|\mathcal{C}_K|$ is bounded by

$$\Gamma^2 |\mathcal{C}_K| \leq 2d \log \left( \frac{\lambda d + |\mathcal{C}_K| L^2}{\lambda d} \right).$$

This result can be re-organized as

$$\frac{\Gamma^2 |\mathcal{C}_K|}{2d} \leq \log \left( 1 + \frac{2L^2}{\Gamma^2 \lambda} \frac{\Gamma^2 |\mathcal{C}_K|}{2d} \right). \tag{D.3}$$

Let $\lambda = B^{-2}$ and since $2L^2 B^2 \geq 2 \geq \Gamma^2$, Lemma D.3 suggests that if

$$\frac{\Gamma^2 |\mathcal{C}_K|}{2d} > 4 \log \left( \frac{4L^2 B^2}{\Gamma^2} \right) + 1 \geq 4 \log \left( \frac{4L^2 B^2}{\Gamma^2} \right) + \frac{\Gamma^2}{2L^2 B^2},$$

then (D.3) will not hold. Thus the necessary condition for (D.3) is

$$\frac{\Gamma^2 |\mathcal{C}_K|}{2d} \leq 4 \log \left( \frac{4L^2 B^2}{\Gamma^2} \right) + 1 = 8 \log \left( \frac{2LB}{\Gamma} \right) + \log(e) = 8 \log \left( \frac{2LBe^{\frac{1}{8}}}{\Gamma} \right) < 8 \log \left( \frac{3LB}{\Gamma} \right).$$

By basic calculus we get the claimed bound for $|\mathcal{C}_K|$ and complete the proof of Lemma C.1. $\qquad \square$

### D.2 PROOF OF LEMMA C.2

The proof follows the standard technique for linear bandits, we first introduce the self-normalized bound for vector-valued martingales from Abbasi-Yadkori et al. (2011).

**Lemma D.4** (Theorem 1, Abbasi-Yadkori et al. 2011). *Let $\{\mathcal{F}_t\}_{t=0}^\infty$ be a filtration. Let $\{\varepsilon_t\}_{t=1}^\infty$ be a real-valued stochastic process such that $\varepsilon_t$ is $\mathcal{F}_t$-measurable and $\varepsilon_t$ is conditionally $R$-sub-Gaussian for some $R \geq 0$. Let $\{\phi_t\}_{t=1}^\infty$ be an $\mathbb{R}^d$-valued stochastic process such that $\phi_t$ is $\mathcal{F}_{t-1}$ measurable and $\|\phi\|_2 \leq L$ for all $t$. For any $t \geq 0$, define $\mathbf{U}_t = \lambda \mathbf{I} + \sum_{k=1}^t \phi_k \phi_k$. Then for any $\delta > 0$, with probability at least $1 - \delta$, for all $t \geq 0$*

$$\left\| \sum_{k=1}^t \phi_k \varepsilon_k \right\|^2_{\mathbf{U}_t^{-1}} \leq 2R^2 \log \left( \frac{\sqrt{\det(\mathbf{U}_t)}}{\sqrt{\det(\mathbf{U}_0)} \delta} \right).$$

**Lemma D.5** (Lemma 8, Zanette et al. (2020c)). *Let $\{\mathbf{a}_i\}_{i=1}^d$ be any sequence of vectors in $\mathbb{R}^d$ and $\{b_i\}_{i=1}^d$ be any sequence of scalars such that $|b_i| \leq \epsilon$. For any $\lambda > 0$:*

$$\left\| \sum_{i=1}^n \mathbf{a}_i b_i \right\|^2_{\left[ \sum_{i=1}^n \mathbf{a}_i \mathbf{a}_i^\top + \lambda \mathbf{I} \right]^{-1}} \leq n \epsilon^2.$$

The next lemma is to bound the perturbation of the misspecification

**Lemma D.6.** *Let $\{\eta_k\}_k$ be any sequence of scalars such that $|\eta_k| \leq \zeta$ for any $k \in [K]$. For any index subset $\mathcal{C} \subseteq [K]$, define $\mathbf{U} = \lambda \mathbf{I} + \sum_{k \in \mathcal{C}} \mathbf{x}_k \mathbf{x}_k^\top$, then for any $\mathbf{x} \in \mathbb{R}^d$, we have*

$$\left| \mathbf{x}^\top \mathbf{U}^{-1} \sum_{k \in \mathcal{C}} \mathbf{x}_k \eta_k \right| \leq \zeta \sqrt{|\mathcal{C}|} \|\mathbf{x}\|_{\mathbf{U}^{-1}}.$$

*Proof.* By Cauchy-Schwartz inequality we have

$$\left| \mathbf{x}^\top \mathbf{U}^{-1} \sum_{k \in \mathcal{C}} \mathbf{x}_k \eta_k \right| \leq \|\mathbf{x}\|_{\mathbf{U}^{-1}} \left\| \sum_{k \in \mathcal{C}} \mathbf{x}_k \eta_k \right\|_{\mathbf{U}^{-1}} \leq \zeta \sqrt{|\mathcal{C}|} \|\mathbf{x}\|_{\mathbf{U}^{-1}},$$

where the second inequality dues to lemma D.5. $\qquad \square$

The next lemma provides the Determinant-Trace inequality.

**Lemma D.7.** Suppose sequence $\{\mathbf{x}_k\}_{k=1}^K \subset \mathbb{R}^d$ and for any $k \in [K]$, $\|\mathbf{x}_k\|_2 \leq L$. For any index subset $\mathcal{C} \subseteq [K]$, define $\mathbf{U} = \lambda\mathbf{I} + \sum_{k \in \mathcal{C}} \mathbf{x}_k\mathbf{x}_k^\top$ for some $\lambda > 0$, then $\det(\mathbf{U}) \leq (\lambda + |\mathcal{C}|L^2/d)^d$.

*Proof.* The proof of this lemma is almost the same with Lemma 10 in Abbasi-Yadkori et al. (2011) by replacing the index set $[K]$ with any subset $\mathcal{C}$. We refer the readers to check Abbasi-Yadkori et al. (2011) for details. $\qquad\square$

Equipped with these lemmas, we can start our proof.

*Proof of Lemma C.2.* For any $k \in [K]$, considering the data samples $k' \in \mathcal{C}_{k-1}$ used for regression at round $k$. Following the update rule of $\mathbf{U}_k$ and $\boldsymbol{\theta}_k$ yields

$$\mathbf{U}_k(\boldsymbol{\theta}_k - \boldsymbol{\theta}^*) = \mathbf{U}_k\mathbf{U}_k^{-1}\left(\sum_{k' \in \mathcal{C}_{k-1}} \mathbf{x}_{k'}r_{k'}\right) - \left(\lambda\mathbf{I} + \sum_{k' \in \mathcal{C}_{k-1}} \mathbf{x}_{k'}\mathbf{x}_{k'}^\top\right)\boldsymbol{\theta}^*$$

$$= \sum_{k' \in \mathcal{C}_{k-1}} \mathbf{x}_{k'}r_{k'} - \lambda\boldsymbol{\theta}^* - \sum_{k' \in \mathcal{C}_{k-1}} \mathbf{x}_{k'}\mathbf{x}_{k'}^\top\boldsymbol{\theta}^*$$

$$= -\lambda\boldsymbol{\theta}^* + \sum_{k' \in \mathcal{C}_{k-1}} \mathbf{x}_{k'}(r_{k'} - \mathbf{x}_{k'}^\top\boldsymbol{\theta}^*)$$

$$= -\lambda\boldsymbol{\theta}^* + \sum_{k' \in \mathcal{C}_{k-1}} \mathbf{x}_{k'}\varepsilon_{k'} + \sum_{k' \in \mathcal{C}_{k-1}} \mathbf{x}_{k'}\eta_{k'},$$

where the first equation is due to the fact that $\mathbf{U}_k = \lambda\mathbf{I} + \sum_{k' \in \mathcal{C}_{k-1}} \mathbf{x}_k\mathbf{x}_k^\top$ and $\boldsymbol{\theta}_k = \mathbf{U}_k^{-1}\sum_{k' \in \mathcal{C}_{k-1}} \mathbf{x}_{k'}r_{k'}$. The last equation follows the fact that $r_{k'}$ is generated from $r_{k'} = r(\mathbf{x}_{k'}) + \varepsilon_{k'} = \mathbf{x}_{k'}^\top\boldsymbol{\theta}^* + \eta(\mathbf{x}_{k'}) + \varepsilon_{k'}$, where we denote $\eta(\mathbf{x}_{k'})$ as $\eta_{k'}$ for the model misspecification error and $\varepsilon_{k'}$ is the random noise. Therefore, consider any contextual vector $\mathbf{x} \in \mathbb{R}^d$, we have

$$\left|\mathbf{x}^\top(\boldsymbol{\theta}_k - \boldsymbol{\theta}^*)\right| = \left|\mathbf{x}^\top\mathbf{U}_k^{-1}\mathbf{U}_k(\boldsymbol{\theta}_k - \boldsymbol{\theta}^*)\right|$$

$$\leq \lambda\underbrace{\left|\mathbf{x}^\top\mathbf{U}_k^{-1}\boldsymbol{\theta}^*\right|}_{q_1} + \underbrace{\left|\mathbf{x}^\top\mathbf{U}_k^{-1}\sum_{k' \in \mathcal{C}_{k-1}} \boldsymbol{\phi}_{k'}\varepsilon_{k'}\right|}_{q_2} + \underbrace{\left|\mathbf{x}^\top\mathbf{U}_k^{-1}\sum_{k' \in \mathcal{C}_{k-1}} \boldsymbol{\phi}_{k'}\eta_{k'}\right|}_{q_3},$$

where the inequality is due to the triangles inequality. Lemma D.6 yields $q_3 \leq \zeta\sqrt{|\mathcal{C}_{k-1}|}\|\mathbf{x}\|_{\mathbf{U}_k^{-1}}$. From the fact that $|\mathbf{x}^\top\mathbf{A}\mathbf{y}| \leq \|\mathbf{x}\|_\mathbf{A}\|\mathbf{y}\|_\mathbf{A}$, we can bound term $q_1$ by

$$q_1 \leq \|\mathbf{x}\|_{\mathbf{U}_k^{-1}}\|\boldsymbol{\theta}^*\|_{\mathbf{U}_k^{-1}} \leq \lambda^{-1/2}B\|\mathbf{x}\|_{\mathbf{U}_k^{-1}}. \tag{D.4}$$

where the last inequality is due to the fact that $\mathbf{U}_i^{-1} \preceq \lambda^{-1}\mathbf{I}$. Term $q_2$ is also bounded as

$$q_2 \leq \|\mathbf{x}\|_{\mathbf{U}_k^{-1}}\left\|\sum_{k' \in \mathcal{C}_{k-1}} \mathbf{x}_{k'}\varepsilon_{k'}\right\|_{\mathbf{U}_k^{-1}} = \|\mathbf{x}\|_{\mathbf{U}_k^{-1}}\underbrace{\left\|\sum_{k'=1}^K \mathbb{1}\left[k' \in \mathcal{C}_{k-1}\right]\mathbf{x}_{k'}\varepsilon_{k'}\right\|_{\mathbf{U}_k^{-1}}}_{I_1}, \tag{D.5}$$

where the second equation uses the indicator to replace the summation over subset $\mathcal{C}_{k-1}$. Denoting $\mathbf{y}_{k'} = \mathbb{1}\left[k' \in \mathcal{C}_{k-1}\right]\mathbf{x}_{k'}$, noticing that $\|\mathbf{y}_k\|_2 \leq \|\mathbf{x}_k\|_2 \leq L$ and

$$\mathbf{U}_k = \sum_{k' \in \mathcal{C}_{k-1}} \mathbf{x}_{k'}\mathbf{x}_{k'}^\top = \sum_{k'=1}^K \mathbb{1}\left[k' \in \mathcal{C}_{k-1}\right]\mathbf{x}_{k'}\mathbf{x}_{k'}^\top = \sum_{k'=1}^K \mathbf{y}_{k'}\mathbf{y}_{k'}^\top,$$

by Lemma D.4, $I_1$ can be further bounded by

$$I_1 \leq \sqrt{2R^2\log\left(\frac{\sqrt{\det(\mathbf{U}_k)}}{\sqrt{\det(\mathbf{U}_0)}\delta}\right)} \leq R\sqrt{2\log\left(\frac{\det(\mathbf{U}_k)}{\det(\mathbf{U}_0)\delta}\right)} = R\sqrt{2\log\left(\frac{\det(\mathbf{U}_k)}{\lambda^d\delta}\right)}, \tag{D.6}$$

where the second inequality follows the fact that $\det(\mathbf{U}_k) \geq \det(\mathbf{U}_0) = \lambda^d$. Notice that $\mathbf{U}_k = \lambda\mathbf{I} + \sum_{k' \in \mathcal{C}_{k-1}} \mathbf{x}_{k'}\mathbf{x}_{k'}^\top$, lemma D.7 suggests that $\det(\mathbf{U}_k) \leq (\lambda + |\mathcal{C}_{k-1}|L^2/d)^d$, plugging this into (D.6) we have $I_1$ can be finally bounded by

$$I_1 \leq R\sqrt{2\log\left(\frac{(\lambda + |\mathcal{C}_{k-1}|L^2/d)^d}{\lambda^d \delta}\right)} \leq R\sqrt{2d\log\left(\frac{d\lambda + |\mathcal{C}_{k-1}|L^2}{d\lambda\delta}\right)}.$$

Plugging the bound of $I_1$ into (D.5) and combining with (D.4) and Lemma D.6 together, replacing $|\mathcal{C}_{k-1}|$ with its upper bound $|\mathcal{C}_K|$ we have with probability at least $1 - \delta$, for all $k \in [K], \mathbf{x} \in \mathbb{R}^d$,

$$|\mathbf{x}^\top(\boldsymbol{\theta}_k - \boldsymbol{\theta}^*)| \leq \left(R\sqrt{2d\log\left(\frac{d\lambda + |\mathcal{C}_K|L^2}{d\lambda\delta}\right)} + B\lambda^{-1/2} + \zeta\sqrt{|\mathcal{C}_K|}\right)\|\boldsymbol{\phi}\|_{\mathbf{U}_k^{-1}}.$$

Letting $\lambda = B^{-2}$ we get the claimed results. $\qquad\square$

## D.3 Proof of Lemma C.4

*Proof.* According to the definition of expected reward function $r(\mathbf{x})$, we have for all $k \in [K]$, suppose the condition in Lemma C.2 holds, then

$$\begin{aligned}
r_k^* - r_k &= \eta(\mathbf{x}_k^*) - \eta(\mathbf{x}_k) + (\mathbf{x}_k^*)^\top \boldsymbol{\theta}^* - \mathbf{x}_k^\top \boldsymbol{\theta}^* \\
&\leq 2\zeta + (\mathbf{x}_k^*)^\top \boldsymbol{\theta}^* - \mathbf{x}_k^\top \boldsymbol{\theta}^* \\
&= 2\zeta + (\mathbf{x}_k^*)^\top \boldsymbol{\theta}_k + (\mathbf{x}_k^*)^\top (\boldsymbol{\theta}^* - \boldsymbol{\theta}_k) - \mathbf{x}_k^\top \boldsymbol{\theta}_k + \mathbf{x}_k^\top (\boldsymbol{\theta}^* - \boldsymbol{\theta}_k) \\
&\leq 2\zeta + (\mathbf{x}_k^*)^\top \boldsymbol{\theta}_k + \beta\|\mathbf{x}_k^*\|_{\mathbf{U}_k^{-1}} - \mathbf{x}_k^\top \boldsymbol{\theta}_k + \beta\|\mathbf{x}_k\|_{\mathbf{U}_k^{-1}} \\
&\leq 2\zeta + \mathbf{x}_k^\top \boldsymbol{\theta}_k + \beta\|\mathbf{x}_k\|_{\mathbf{U}_k^{-1}} - \mathbf{x}_k^\top \boldsymbol{\theta}_k + \beta\|\mathbf{x}_k\|_{\mathbf{U}_k^{-1}} \\
&\leq 2\zeta + 2\beta\|\mathbf{x}_k\|_{\mathbf{U}_k^{-1}},
\end{aligned}$$

where the second inequality utilize the fact that $|\eta(\mathbf{x})| \leq \zeta$ for all $\mathbf{x} \in \mathcal{D}_k$. The inequality on the forth line follows Corollary C.3. The inequality on the fifth line is due to the fact that $\mathbf{x}_k = \arg\max_{\mathbf{x} \in \mathcal{D}_k} \mathbf{x}^\top \boldsymbol{\theta}_k + \beta\|\mathbf{x}\|_{\mathbf{U}_k^{-1}}$, which is executed in Line 4 in Algorithm 1. $\qquad\square$

## D.4 Proof of Lemma C.6

*Proof.* First it is clear to see that $\sqrt{2\iota_3} = \sqrt{2\log(1 + 16L^2B^2\Gamma^{-2}\iota_2) + 2\log(1/\delta)}$. Using the fact that $\sqrt{a + b} \leq \sqrt{a} + \sqrt{b}$, it can be further bounded by

$$\sqrt{2\iota_3} \leq \sqrt{2\log(1 + 16L^2B^2\Gamma^{-2}\iota_2)} + \sqrt{2\log(1/\delta)}.$$

Assuming $L \geq 1, B \geq 1, \Gamma = \Delta/(2\sqrt{d}\iota_1) \leq 1$ yields $LB\Gamma^{-1} \geq 1$, then by basic calculus one can verify that

$$2 + 4\sqrt{\iota_2} \leq 6\log(3LB\Gamma^{-1}), \quad \sqrt{2\log(1 + 16L^2B^2\Gamma^{-2}\iota_2)} \leq 3\log(3LB\Gamma^{-1}),$$

therefore we have that

$$\begin{aligned}
2 + 4\sqrt{\iota_2} + R\sqrt{2\iota_3} &\leq (6 + 3R)\log(3LB\Gamma^{-1}) + \sqrt{2\log(1/\delta)}R \\
&= (6 + 3R)\log(6LB\sqrt{d}\Delta^{-1}\iota_1) + \sqrt{2\log(1/\delta)}R,
\end{aligned}$$

where the last equality is from the fact that $\Gamma = \Delta/(2\sqrt{d}\iota_1)$. Lemma D.2 suggests that the necessary condition for

$$\underbrace{(6LB\sqrt{d}\Delta^{-1})\iota_1}_{x} \geq \underbrace{(6LB\sqrt{d}\Delta^{-1})(6 + 3R)}_{a}\log(6LB\sqrt{d}\Delta^{-1}\iota_1) + \underbrace{(6LB\sqrt{d}\Delta^{-1})\sqrt{2\log(1/\delta)}R}_{b}$$

$$(\text{D.7})$$

is that

$$(6LB\sqrt{d}\Delta^{-1})\iota_1 \geq 4(6LB\sqrt{d}\Delta^{-1})(6+3R)\log(2(6LB\sqrt{d}\Delta^{-1})(6+3R))$$
$$+ 2(6LB\sqrt{d}\Delta^{-1})\sqrt{2\log(1/\delta)}R,$$

which suggests that setting

$$\iota_1 = (24+18R)\log((72+54R)LB\sqrt{d}\Delta^{-1}) + \sqrt{8R^2\log(1/\delta)}$$

implies the fact that $\iota_1 \geq 2 + 4\sqrt{\iota_2} + R\sqrt{2\iota_3}$ $\qquad\square$

## E   PROOF OF THEOREM 5.4

To begin with, we introduce the lemma providing a sparse vector set in $\mathbb{R}^d$.

**Lemma E.1** (Lemma 3.1, Lattimore et al. 2020). For any $\varepsilon > 0$ and $d < [|\mathcal{D}|]$ such that $d \geq \lceil 8\log(|\mathcal{D}|)\varepsilon^{-2}\rceil$, there exists a vector set $\mathcal{D} \subset \mathbb{R}^d$ such that $\|\mathbf{x}\|_2 = 1$ for all $\mathbf{x} \in \mathcal{D}$ and $|\langle \mathbf{x}, \mathbf{y}\rangle| \leq \varepsilon$ for all $\mathbf{x}, \mathbf{y} \in \mathcal{D}$ and $\mathbf{x} \neq \mathbf{y}$.

Next, we present the Bretagnolle–Huber inequality providing the lower bound to distinguish a system.

**Lemma E.2** (Bretagnolle–Huber inequality). Let $P$ and $Q$ be probability measures on the same measurable space $(\Omega, \mathcal{F})$, let $\mathcal{A} \in \mathcal{F}$ be an arbitary event. Then

$$P(\mathcal{A}) + Q(\mathcal{A}^c) \geq \frac{1}{2}\exp(-\text{KL}(P,Q)).$$

For stochastic linear bandit problem with finite arm, we can denote $T_i(k)$ as the number of rounds the algorithm visit the $i$-th arm over total $k$ rounds. Then We have the KL-divergence decomposition lemma.

**Lemma E.3** (Lemma 15.1, Lattimore & Szepesvári (2020)). Let $\nu = (P_1, \cdots, P_n)$ be the reward distributions associated with one $n$-armed bandit and let $\nu' = (P_1', \cdots, P_n')$ be another $n$-armed bandit. Fix some algorithm $\pi$ and let $\mathbb{P}_\nu = \mathbb{P}_{\nu\pi}, \mathbb{P}_{\nu'} = \mathbb{P}_{\nu',\pi}$ be the probability measures on the canonical bandit model induced by the $k$-round interconnection of $\pi$ and $\nu$ (respectively, $\pi$ and $\nu'$). Then $\text{KL}(\mathbb{P}_\nu, \mathbb{P}_{\nu'}) = \sum_{i=1}^n \mathbb{E}_\nu[T_i(n)]\text{KL}(P_i, P_i')$

*Proof of Theorem 5.4.* The proof starts from inheriting the idea from Lattimore et al. (2020). Given dimension $d$ and the number of arms $|\mathcal{D}|$, setting $\varepsilon = \sqrt{8\log(|\mathcal{D}|)/(d-1)}$, we can provide the contextual vector set $\mathcal{D}$ such that

$$\|\mathbf{x}\|_2 = 1, \forall \mathbf{x} \in \mathcal{D}, |\langle \mathbf{x}, \mathbf{y}\rangle| \leq \sqrt{\frac{8\log(|\mathcal{D}|)}{d-1}}, \forall \mathbf{x}, \mathbf{y} \in \mathcal{D}, \mathbf{x} \neq \mathbf{y},$$

For simplicity, we index the decision set as $\mathbf{x}_1, \cdots, \mathbf{x}_{|\mathcal{D}|}$. Given the minimal sub-optimality gap $\Delta$, we provide the parameter set $\boldsymbol{\Theta}$ as follows:

$$\boldsymbol{\Theta} = \left\{\boldsymbol{\theta}_{(i,j)} = \Delta\mathbf{x}_i + 2\Delta\mathbf{x}_j, \mathbf{x}_i, \mathbf{x}_j \in \mathcal{D}, i \neq j\right\}\bigcup\{\boldsymbol{\theta}_i = \Delta\mathbf{x}_i, \mathbf{x}_i \in \mathcal{D}\}.$$

It can be verified that $\boldsymbol{\Theta}$ contains two kinds of $\boldsymbol{\theta}$. The first one $\boldsymbol{\theta}_{(i,j)}$ is a mixture of two different contexts $\mathbf{x}_i, \mathbf{x}_j$ with different strength $\Delta$ and $2\Delta$. The second one is $\boldsymbol{\theta}_i$ which only contains features from one context $\mathbf{x}_i$. We can further verify that the size of $|\boldsymbol{\Theta}| = |\mathcal{D}|^2$ and $\|\boldsymbol{\theta}\|_2 \leq \sqrt{5}\Delta$ for $\boldsymbol{\theta} \in \boldsymbol{\Theta}$. For different parameter $\boldsymbol{\theta}$, the reward function is sampled from a Gaussian distribution $\mathcal{N}(r_{\boldsymbol{\theta}}(\mathbf{x}), 1)$, where the expected reward function is defined as

$$r_{\boldsymbol{\theta}_{(i,j)}}(\mathbf{x}) = \begin{cases} 2\Delta \text{ if } \mathbf{x} = \mathbf{x}_j \\ \Delta \text{ if } \mathbf{x} = \mathbf{x}_i \\ 0 \text{ otherwise} \end{cases}, r_{\boldsymbol{\theta}_i}(\mathbf{x}) = \begin{cases} \Delta \text{ if } \mathbf{x} = \mathbf{x}_i \\ 0 \text{ otherwise} \end{cases}.$$

We can verify that the minimal sub-optimality of all these bandit problem is $\Delta$. For different parameter $\boldsymbol{\theta}$ and input $\mathbf{x}$, by utilizing the sparsity of the set $\mathcal{D}$ (i.e. $|\mathbf{x}^\top y| \leq \varepsilon$ if $\mathbf{x} \neq \mathbf{y}$), we can verify the misspecification level as

$$|r_{\boldsymbol{\theta}_{(i,j)}}(\mathbf{x}) - \boldsymbol{\theta}_{(i,j)}^\top \mathbf{x}| = \begin{cases} |2\Delta - 2\Delta\mathbf{x}_j^\top \mathbf{x} - \Delta\mathbf{x}_i^\top \mathbf{x}| \leq \Delta\varepsilon & \text{if } \mathbf{x} = \mathbf{x}_j \\ |\Delta - 2\Delta\mathbf{x}_j^\top \mathbf{x} - \Delta\mathbf{x}_i^\top \mathbf{x}| \leq 2\Delta\epsilon & \text{if } \mathbf{x} = \mathbf{x}_i \\ |0 - 2\Delta\mathbf{x}_j^\top \mathbf{x} - \Delta\mathbf{x}_i^\top \mathbf{x}| \leq 3\Delta\varepsilon & \text{otherwise} \end{cases}$$

$$|r_{\boldsymbol{\theta}_i}(\mathbf{x}) - \boldsymbol{\theta}_i^\top(\mathbf{x})| = \begin{cases} |\Delta - \Delta\mathbf{x}_i^\top \mathbf{x}| = 0 & \text{if } \mathbf{x} = \mathbf{x}_i \\ |0 - \Delta\mathbf{x}_i^\top \mathbf{x}| \leq \Delta\varepsilon & \text{otherwise.} \end{cases}$$

Therefore we have verified that the misspecification level is bounded by $\zeta = 3\Delta\varepsilon$.

The provided bandit structure is hard for any linear algorithm to learn since any algorithm cannot get any information before it encounters non-zero expected rewards, even regardless of the noise of the rewards. We following the same method in Lattimore & Szepesvári (2020). If the algorithm choose arm $i$ at the first round, there would be $|\mathcal{D}|$ parameters (i.e. $\boldsymbol{\theta}_i, \boldsymbol{\theta}_{(i,\cdot)}$ receiving a non-zero expected reward. On the second round if the algorithm choose a different arm $j$, there would be $|\mathcal{D}|$ parameters (i.e. $\boldsymbol{\theta}_j, \boldsymbol{\theta}_{(j,k:k\neq i)}$ receiving a non-zero expected reward. Therefore the average time of receiving zero expected reward should be

$$|\mathcal{D}|^{-2} \sum_{i=1}^{|\mathcal{D}|} (i-1)(|\mathcal{D}| - i + 1) = |\mathcal{D}|^{-2} \sum_{i=0}^{|\mathcal{D}|-1} i(|\mathcal{D}| - i)$$

$$= |\mathcal{D}|^{-2} \left( |\mathcal{D}| \sum_{i=0}^{|\mathcal{D}|-1} i - \sum_{i=0}^{|\mathcal{D}|-1} i^2 \right)$$

$$= |\mathcal{D}|^{-2} \left( \frac{|\mathcal{D}|^2(|\mathcal{D}| - 1)}{2} - \frac{|\mathcal{D}|(|\mathcal{D}| - 1)(2|\mathcal{D}| - 1)}{6} \right)$$

$$= \frac{|\mathcal{D}| - 1}{2} \left( 1 - \frac{2|\mathcal{D}| - 1}{3|\mathcal{D}|} \right)$$

$$\geq \frac{|\mathcal{D}| - 1}{6},$$

where the third equation is from the fact that $\sum_{i=1}^n i = n(n+1)/2$ and $\sum_{i=1}^n i^2 = n(n+1)(2n+1)/6$. The last inequality is from the fact that $2|\mathcal{D}| - 1)/(3|\mathcal{D}|) \leq 2/3$. Therefore, even without of the random noise, any algorithm is expected to receive $\min\{K, (|\mathcal{D}| - 1)/6\}$ uninformative data with expected reward to be zero. Therefore any algorithm will receive a $\Delta \min\{K, (|\mathcal{D}| - 1)/6\}$ regret considers the suboptimality as $\Delta$.

Next, we consider the effect of random noise. For any algorithm running on this parameter set $\boldsymbol{\Theta}$, we find two parameter $\boldsymbol{\theta}_i$ and $\boldsymbol{\theta}_{i,j}$ where $j \neq i$. Define the event as $\mathcal{A} = \{T_j(k) \geq k/2\}$ and $\mathcal{A}^c = \{T_j(k) < k/2\}$. By Lemma E.2 and Lemma E.3,

$$\mathbb{P}_{\boldsymbol{\theta}_i}\left(T_j(k) \geq \frac{k}{2}\right) + \mathbb{P}_{\boldsymbol{\theta}_{(i,j)}}\left(T_j(k) < \frac{k}{2}\right) \geq \frac{1}{2}\exp(-\mathrm{KL}(\mathbb{P}_{\boldsymbol{\theta}_i}, \mathbb{P}_{\boldsymbol{\theta}_{(i,j)}}))$$

$$\geq \frac{1}{2}\exp\left(-\sum_{n\in\mathcal{D}} \mathbb{E}_{\boldsymbol{\theta}_i}[T_n(k)]\mathrm{KL}\left(\mathbb{P}_{\boldsymbol{\theta}_{(i,j)},n}, \mathbb{P}_{\boldsymbol{\theta}_j,n}\right)\right).$$

$$\text{(E.1)}$$

Noticing the minimal sub-optimality gap is $\Delta$. Also the $j$-th arm is the sub-optimal arm for parameter $\boldsymbol{\theta}_i$. Therefore, once $T_j(k) \geq k/2$, the algorithm will at least suffer from $\Delta k/2$ regret for parameter $\boldsymbol{\theta}_i$. Also, since the $j$-th arm is the optimal arm for bandit $\boldsymbol{\theta}_{(i,j)}$. If $T_j(k) < k/2$, the algorithm will also at least suffer from $\Delta k/2$ regret for $\boldsymbol{\theta}_{(i,j)}$. Denoting $\mathcal{R}_{\boldsymbol{\theta}}(k)$ as the expected cumulative regret over $k$ rounds, that is to say

$$\mathcal{R}_{\boldsymbol{\theta}_i}(k) \geq \frac{\Delta k}{2}\mathbb{P}_{\boldsymbol{\theta}_i}(T_j(k) \geq k/2) \quad \mathcal{R}_{\boldsymbol{\theta}_j}(k) \geq \frac{\Delta k}{2}\mathbb{P}_{\boldsymbol{\theta}_i}(T_j(k) < k/2). \quad \text{(E.2)}$$

On the other hand since the bandit using $\boldsymbol{\theta}_i$ and $\boldsymbol{\theta}_j$ only differ in the $j$-th arm. Since standard Gaussian noise is adapted, $\mathrm{KL}(\mathbb{P}_{\boldsymbol{\theta}_i,n}, \mathbb{P}_{\boldsymbol{\theta}_{(i,j)},n}) = \Delta^2 \mathbb{1}[n = j]/2$. Combining this with (E.2), (E.1) suggests that

$$\mathcal{R}_{\boldsymbol{\theta}_i}(k) + \mathcal{R}_{\boldsymbol{\theta}_j}(k) \geq \frac{\Delta k}{2} \exp\left(-\frac{\Delta^2}{2}\mathbb{E}_{\boldsymbol{\theta}_i}\left[T_j(k)\right]\right),$$

which suggests that

$$\mathbb{E}_{\boldsymbol{\theta}_i}\left[T_j(k)\right] \geq \frac{\log(\Delta k) - \log 2 - \log(\mathcal{R}_{\boldsymbol{\theta}_i}(k) + \mathcal{R}_{\boldsymbol{\theta}_j}(k))}{\Delta^2/2}, \tag{E.3}$$

For any algorithm seeking to get a sublinear expected regret bound of $\mathcal{R}_{\boldsymbol{\theta}}(k) \leq Ck^\alpha$ with $C > 0, 0 \leq \alpha < 1$ for all $\boldsymbol{\theta} \in \boldsymbol{\Theta}$, (E.3) becomes

$$\mathbb{E}_{\boldsymbol{\theta}_i}\left[T_j(k)\right] \geq \frac{\log(\Delta k) - \log 2 - \log(2Ck^\alpha)}{\Delta^2/2} = \frac{\log(\Delta k) - \log(4C) - \alpha \log k}{\Delta^2/2}. \tag{E.4}$$

Since that the regret on $\boldsymbol{\theta}_i$ can be decomposed by

$$\mathcal{R}_{\boldsymbol{\theta}_i}(k) = \Delta \sum_{n=1, n \neq i}^{|\mathcal{D}|} T_n(k), \tag{E.5}$$

combining (E.5) with (E.4) yields

$$\mathcal{R}_{\boldsymbol{\theta}_i}(k) \geq \frac{2(|\mathcal{D}| - 1)}{\Delta} \max\left\{\log(\Delta k) - \log(4C) - \alpha \log k, 0\right\},$$

where the $\max$ operator is trivially taken for $\mathcal{R}_{\boldsymbol{\theta}}(k) \geq 0$.

$\square$

# F    PROOF OF THEOREM 7.4

In this section we provide the proof of Theorem 7.4, we start from the technical lemmas for the proof. The first lemma is similar with Lemma C.1 by setting $\lambda = L = 1$ as in Definition 7.1 and taking union for $H$ possible cases

**Lemma F.1.** Given $0 < \Gamma \leq 1$. For any $k \in [K]$, $|\mathcal{C}_k| \leq 8Hd\Gamma^{-2}\log(6H\Gamma^{-2})$.

Then the next lemma extends Lemma C.2 and Lemma C.4 to $H > 1$ setting and is similar with Lemma C.5 in Jin et al. (2020), by replacing the total number $K$ with $|\mathcal{C}_K|$ used in regression

**Lemma F.2** (Lemma C.5, Jin et al. 2020)**.** Let $\widetilde{\beta} = c_\beta H(d\sqrt{\iota_2} + \zeta\Gamma^{-1}\sqrt{8Hd\iota_1})$, where $\iota_1 = \log(6H\Gamma^{-2})$, $\iota_2 = \log((16H^2d^2\Gamma^{-2}\iota_1)/\delta)$ and $c_\beta$ is an absolute constant. With probability at least $1 - \delta$, for any fixed policy $\pi$ and $(s, a, h, k) \in \mathcal{S} \times \mathcal{A} \times [H] \times [K]$,

$$\boldsymbol{\phi}^\top(s, a)\mathbf{w}_h^k - Q_h^\pi(s, a) = P_h(V_{h+1}^k - V_{h+1}^\pi)(s, a) + \rho_h^k(s, a),$$

where $|\rho_h^k(s, a)| \leq \widetilde{\beta}\|\boldsymbol{\phi}(s, a)\|_{(\mathbf{U}_h^k)^{-1}} + 4H\zeta$.

Given Lemma F.2 we can provide the rate of confidence radius $\beta$ by adapting the condition and parameters setting: on the misspecification level and the parameter setting

**Corollary F.3.** Let the misspecification level be bounded by $\zeta = \widetilde{\mathcal{O}}(\Delta d^{-0.5}H^{-2.5})$ and choose the threshold parameter to be $\Gamma = \widetilde{\Theta}(\Delta d^{-1}H^{-2})$, one can verify that the $\widetilde{\beta}$ in Lemma F.2 can be bounded by $\widetilde{\beta} \leq \beta = \widetilde{\mathcal{O}}(Hd)$.

Given this corollary, we show that the sub-optimality gap is controlled by three parts: summation of the UCB bonus, misspecification level and the noise induced by the probability transition kernel.

**Lemma F.4.** Suppose Corollary F.3 and Lemma F.2 holds, for any subset $\mathcal{K} \subseteq [K]$ and $h \in [H]$ we have

$$\sum_{k \in \mathcal{K}} \Delta_h(s_h^k, a_h^k) \leq \sum_{k \in \mathcal{K}} V_h^*(s_h^k) - V_h^{\pi^k}(s_h^k)$$

$$\leq 2\beta \sum_{k \in \mathcal{K}} \sum_{h'=h}^H \|\phi_{h'}^k\|_{(\mathbf{U}_{h'}^k)^{-1}} + 4H^2 |\mathcal{K}| \zeta + \sum_{k \in \mathcal{K}} \sum_{h'=h+1}^H \varepsilon_{h'}^k,$$

where $\varepsilon_h^k := [P_h(V_{h+1}^k - V_{h+1}^{\pi^k})](s_h^k, a_h^k) - (V_{h+1}^k(s_{h+1}^k) - V_{h+1}^{\pi^k}(s_{h+1}^k))$ is a zero-mean random variable conditioned on all randomness before episode $k$.

Then the next lemma suggests that the cumulative regret can be bounded by the summation of sub-optimality gap at all stage, with high probability

**Lemma F.5** (Lemma 6.1, He et al. 2021a). For each MDP $\mathcal{M}(\mathcal{S}, \mathcal{A}, H, \{r_h\}, \{P_h\})$, with probability at least $1 - \delta$, the cumulative regret over $K$ episode is bounded by

$$\text{Regret}(K) \leq 2 \sum_{k=1}^K \sum_{h=1}^H \Delta_h(s_h^k, a_h^k) + \frac{16H^3}{3} \log\left(\frac{\log(HK)+1}{\delta}\right) + 2.$$

Equipped with these lemmas, we can start our proof.

*Proof of Theorem 7.4.* For simplicity, we denote the non-selected episode set as $\widetilde{\mathcal{C}}_K := [K] \setminus \mathcal{C}_K$. We first assume Lemma F.2 holds. We fix stage $h$ and then define the following sequence to apply the peeling technique which is also used in He et al. (2021a). For $0 \leq l \leq \log(H/\Delta)/\log(2), l \in \mathbb{N}$, let $k_0^l = 0$ and

$$k_i^l = \min\{k : k > k_{i-1}^l, 2^l \Delta \leq \Delta_h(s_h^k, a_h^k) < 2^{l+1}\Delta, k \in [K]\}.$$

Since $\Delta_h(s, a) \leq H$, there exists $1 + \log(H/\Delta)/\log(2)$ levels. We further denote $\mathcal{K}_l = \{k_1^l, \cdots, k_{|\mathcal{K}_l|}^l\}$ to be the set of the sequence $\{k_i^l\}_i$. We fix one level $l$ in the following proof. On the one hand, due to the fact that $\Delta_h(s_h^k, a_h^k)$ is lower bounded, we have

$$\sum_{k \in \mathcal{K}_l} \Delta_h(s_h^k, a_h^k) \geq 2^l \Delta |\mathcal{K}_l|. \tag{F.1}$$

On the other hand, Lemma F.4 yields

$$\sum_{k \in \mathcal{K}_l} \Delta_h(s_h^k, a_h^k) \leq 2\beta \underbrace{\sum_{k \in \mathcal{K}_l} \sum_{h'=h}^H \|\phi_{h'}^k\|_{(\mathbf{U}_{h'}^k)^{-1}}}_{I_1^l} + 4H^2 |\mathcal{K}_l| \zeta + \underbrace{\sum_{k \in \mathcal{K}_l} \sum_{h'=h}^H \varepsilon_{h'}^k}_{I_2^l}. \tag{F.2}$$

By Azuma-Hoeffding's inequality, with probability at least $1 - \delta/H$, $I_2^l$ is bounded by

$$I_2^l \leq \sqrt{2|\mathcal{K}_l|H^3 \log\left(\frac{HK}{\delta}\right)}. \tag{F.3}$$

Since $[K] = \widetilde{\mathcal{C}}_K \cup \mathcal{C}_K$, term $I_1^l$ is be decomposed as

$$I_1^l = \sum_{k \in \mathcal{K}_l} \sum_{h'=h}^H \|\phi_{h'}^k\|_{(\mathbf{U}_{h'}^k)^{-1}} = \sum_{k \in \mathcal{K}_l \cap \widetilde{\mathcal{C}}_K} \sum_{h'=h}^H \|\phi_{h'}^k\|_{(\mathbf{U}_{h'}^k)^{-1}} + \sum_{k \in \mathcal{K}_l \cap \mathcal{C}_K} \sum_{h'=h}^H \|\phi_{h'}^k\|_{(\mathbf{U}_{h'}^k)^{-1}}$$

$$\leq H\Gamma |\mathcal{K}_l \cap \widetilde{\mathcal{C}}_K| + H\sqrt{|\mathcal{K}_l \cap \mathcal{C}_K|} \sqrt{\sum_{k \in \mathcal{K}_l \cap \mathcal{C}_K} \|\phi_{h'}^k\|_{(\mathbf{U}_{h'}^k)^{-1}}^2}$$

$$\leq H\Gamma |\mathcal{K}_l \cap \widetilde{\mathcal{C}}_K| + H\sqrt{|\mathcal{K}_l \cap \mathcal{C}_K|} \sqrt{2d\log(1 + |\mathcal{K}_l \cap \mathcal{C}_K|)},$$

$$\leq H\Gamma |\mathcal{K}_l| + H\sqrt{|\mathcal{K}_l|} \sqrt{2d\log(1 + K)}, \tag{F.4}$$

where the first term in the first inequality utilizes that $\|\phi_h^k\| \leq \Gamma$ for all $h$ if $k \in \widetilde{C}_K$ and the second term is from triangle's inequality. The second inequality utilizes Lemma C.5. The third inequality is because both $|\mathcal{K}_l \cap \mathcal{C}_K|$ and $|\mathcal{K}_l \cap \widetilde{\mathcal{C}}_K|$ is smaller than $|\mathcal{K}_l|$ as well as $K$. Plugging (F.3) and (F.4) into (F.2) then combining (F.2) with (F.1) yields

$$2^l \Delta |\mathcal{K}_l| \leq \underbrace{(2\beta H \Gamma + 4H^2\zeta)}_{I_3} |\mathcal{K}_l| + \beta H \sqrt{8|\mathcal{K}_l| d \log(1+K)} + \sqrt{2|\mathcal{K}_l| H^3 \log\left(\frac{HK}{\delta}\right)}. \quad \text{(F.5)}$$

Recall the parameter setting $\beta = \widetilde{\mathcal{O}}(Hd)$, $\Gamma = \widetilde{\mathcal{O}}(\Delta d^{-0.5} H^{-2})$, as long as $\zeta = \widetilde{\mathcal{O}}(\Delta d^{-1} H^{-2.5})$, it can be guaranteed that $I_3 \leq \Delta/2 \leq 2^l \Delta/2$. The calculation of the logarithmic terms can follow the proof of Theorem 5.1 where we ignore it for simplicity. Plugging this into (F.5) we have that when the condition on $\zeta$ is satisfied,

$$2^l \Delta |\mathcal{K}_l| \leq \beta H \sqrt{32|\mathcal{K}_l| d \log(1+K)} + \sqrt{8|\mathcal{K}_l| H^3 \log\left(\frac{HK}{\delta}\right)},$$

by the fact that $(\sqrt{a} + \sqrt{b})^2 \leq 2a + 2b$, this implies

$$4^l \Delta^2 |\mathcal{K}_l| \leq 64\beta^2 H^2 d \log(1+K) + 16H^3 \log\left(\frac{HK}{\delta}\right) := I_4. \quad \text{(F.6)}$$

This suggests that $|\mathcal{K}_l| \leq 4^{-l} \Delta^{-2} I_4$ with probability at least $1 - \delta/H$. Since for all $k \in \mathcal{K}_l$, $\Delta_h(s_h^k, a_h^k) < 2^{l+1}\Delta$, the cumulative sub-optimality gap in set $\mathcal{K}_l$ is bounded by

$$\sum_{k \in \mathcal{K}_l} \Delta_h(s_h^k, a_h^k) \leq 2^{l+1}\Delta \times 4^{-l} \Delta^{-2} I_4 \leq 2 \times 2^{-l} \Delta^{-1} I_4. \quad \text{(F.7)}$$

Replacing $\delta$ with $\delta/(1 + \log(H\Delta^{-1})/\log(2))$, we have that with probability at least $1 - \delta/H$, (F.7) holds for all $0 \leq l \leq \log(H\Delta^{-1})/\log(2)$, $l \in \mathbb{N}$ by union bound. Therefore we have that

$$\sum_{l=0}^{\lfloor \log(H/\Delta)/\log(2) \rfloor} \sum_{k \in \mathcal{K}_l} \Delta_h(s_h^k, a_h^k) \leq 2 \sum_{l=0}^{\infty} 2^{-l} \Delta^{-1} I_4 = 4\Delta^{-1} I_4. \quad \text{(F.8)}$$

By union bound we have with probability at least $1 - \delta$, (F.8) holds for all $h \in [H]$ thus

$$\sum_{k=1}^{K} \sum_{h=1}^{H} \Delta_h(s_h^k, a_h^k) \leq 4H\Delta^{-1} I_4. \quad \text{(F.9)}$$

By Lemma F.5, plugging the value of $I_4$ in (F.6) back to (F.9) yields

$$\begin{aligned}
\text{Regret}(K) &\leq 2 \sum_{k=1}^{K} \sum_{h=1}^{H} \Delta_h(s_h^k, a_h^k) + \frac{16H^3}{3} \log\left(\frac{\log(HK)+1}{\delta}\right) + 2 \\
&\leq 8H\Delta^{-1} I_4 + \frac{16H^3}{3} \log\left(\frac{\log(HK)+1}{\delta}\right) + 2 \\
&= 512\beta^2 H^3 d\Delta^{-1} \log(1+K) + 128H^4 \Delta^{-1} \log(HK/\delta) \\
&\quad + \frac{16H^3}{3} \log\left(\frac{\log(HK)+1}{\delta}\right) + 2,
\end{aligned}$$

with probability at least $1 - 3\delta$ by union bound over the probability event in Lemma F.2, Lemma F.5 and (F.9). By Corollary F.3, $\beta = \widetilde{\mathcal{O}}(Hd + \sqrt{H^3 d\zeta \Gamma^{-1}}) = \widetilde{\mathcal{O}}(Hd)$, replace $\delta$ with $\delta/3$, we have the regret is bounded by

$$\text{Regret}(K) \leq \widetilde{\mathcal{O}}(H^5 d^3 \Delta^{-1} \log(K))$$

with probability at least $1 - \delta$. $\qquad \square$

# G    PROOF OF LEMMAS IN APPENDIX F

## G.1    PROOF OF LEMMA F.1

The proof technique is similar with Lemma C.1 equipped with Lemma D.1 and the Lemma D.3.

*Proof of Lemma F.1.* Since the selected data samples follows that there exists an $h \in [H]$ such that $\|\phi_h^k\|_{(\mathbf{U}_h^k)^{-1}} \geq \Gamma$, therefore the summation of the data is lower bounded by

$$\sum_{i \in \mathcal{C}_k} \sum_{h=1}^{H} \min \left\{ 1, \|\phi_h^i\|_{(\mathbf{U}_h^i)^{-1}}^2 \right\} \geq |\mathcal{C}_k| \Gamma^2, \tag{G.1}$$

on the other hand, Lemma D.1 yields

$$\sum_{i \in \mathcal{C}_k} \sum_{h=1}^{H} \min \left\{ 1, \|\phi_h^i\|_{(\mathbf{U}_h^i)^{-1}}^2 \right\} \leq 2dH \log(1 + |\mathcal{C}_k|/d), \tag{G.2}$$

where $\lambda = L = 1$. Combining (G.1) with (G.2) yields

$$\Gamma^2 |\mathcal{C}_k| \leq 2dH \log(1 + |\mathcal{C}_k|/d), \tag{G.3}$$

then following the same rule as the proof of Lemma C.1 we have the necessary condition for (G.3) is

$$|\mathcal{C}_k| \leq 8Hd\Gamma^{-2} \log(6H\Gamma^{-2}). \tag{G.4}$$

$\square$

## G.2    PROOF OF LEMMA F.2

*Proof of Lemma F.2.* The proof of this lemma follows the proof of Lemma C.5 in Jin et al. (2020), from which we have

$$\mathbf{w}_h^k - \mathbf{w}_h^\pi = \underbrace{-\lambda(\mathbf{U}_h^k)^{-1} \mathbf{w}_h^\pi}_{\mathbf{q}_1} + \underbrace{(\mathbf{U}_h^k)^{-1} \sum_{t \in \mathcal{C}_{k-1}} \phi(s_h^t)[V_{h+1}^t(s_{h+1}^t) - [P_h V_{h+1}^t](s_h^t, a_h^t)]}_{\mathbf{q}_2}$$

$$+ \underbrace{(\mathbf{U}_h^k)^{-1} \sum_{t \in \mathcal{C}_{k-1}} \phi(s_h^t, a_h^t)[\widetilde{P}_h(V_{h+1}^t - V_{h+1}^\pi)](s_h^t, a_h^t)}_{\mathbf{q}_3}$$

$$+ \underbrace{(\mathbf{U}_h^k)^{-1} \sum_{t \in \mathcal{C}_{k-1}} \phi(s_h^t, a_h^t)[r_h^t - \langle \phi^\top(s_h^t, a_h^t)\boldsymbol{\theta}_h \rangle + [(P_h - \widetilde{P}_h)V_{h+1}^t](s_h^t, a_h^t)}_{\mathbf{q}_4}$$

where $\widetilde{P}(\cdot|s,a)$ is the well-specified transition kernel defined by $\widetilde{P}(\cdot|s,a) = \langle \boldsymbol{\mu}(\cdot), \phi(s,a) \rangle$. From the proof in Jin et al. (2020) we have that for any $(s,a) \in \mathcal{S} \times \mathcal{A}$,

$$|\langle \phi(s,a), \mathbf{q}_1 \rangle| \leq \sqrt{\lambda} \|\mathbf{w}_h^\pi\|_2 \|\phi(s,a)\|_{(\mathbf{U}_h^k)^{-1}} \leq \sqrt{\lambda} B \|\phi(s,a)\|_{(\mathbf{U}_h^k)^{-1}}$$

$$|\langle \phi(s,a), \mathbf{q}_2 \rangle| \leq \widetilde{\mathcal{O}}(dH) \|\phi(s,a)\|_{(\mathbf{U}_h^k)^{-1}}$$

$$\langle \phi(s,a), \mathbf{q}_3 \rangle = \widetilde{P}_h(V_{h+1}^k - V_{h+1}^\pi)(s,a) + p_2, |p_2| \leq 2H\sqrt{d\lambda} \|\phi(s,a)\|_{(\mathbf{U}_h^k)^{-1}}.$$

For the fourth term, by Lemma D.6, which improves the Lemma C.4 in Jin et al. (2020) by a factor of $\sqrt{d}$, we have

$$|\langle \phi(s,a), \mathbf{q}_4 \rangle| \leq 2H\zeta\sqrt{|\mathcal{C}_{k-1}|} \|\phi(s,a)\|_{(\mathbf{U}_h^k)^{-1}}.$$

Finally, since $\langle \phi(s,a), \mathbf{w}_h^k - \mathbf{w}_h^\pi \rangle = \langle \phi(s,a), \mathbf{q}_1 + \mathbf{q}_2 + \mathbf{q}_3 + \mathbf{q}_4 \rangle$ we have that

$$|\langle \phi(s,a), \mathbf{w}_h^k \rangle - Q_h^\pi(s,a) - [P_h(V_{h+1}^k - V_{h+1}^\pi)](s,a)| \leq \widetilde{\mathcal{O}}(Hd + H\zeta\sqrt{|\mathcal{C}_{k-1}|}) \|\phi(s,a)\|_{(\mathbf{U}_h^k)^{-1}},$$

plugging the bound of $|\mathcal{C}_{k-1}|$ in Lemma F.1 back we will have the claimed results.    $\square$

## G.3   PROOF OF LEMMA F.4

The proof of Lemma F.4 follows the same technique from Jin et al. (2020) and we warp it here for completeness. The first lemma shows that the estimation of $Q$ function is still optimistic regardless the misspecification.

**Lemma G.1** (Lemma C.5, Jin et al. 2020). Under the parameter setting in Theorem 7.4 and Lemma F.2 holds, we have $Q_h^k(s,a) \geq Q_h^*(s,a) - 4H(H+1-h)\zeta$ for all $(s,a,h,k) \in \mathcal{S} \times \mathcal{A} \times [H] \times [K]$.

Next lemma provides the recursive error bound for the estimated value function $V_h^k(\cdot)$

**Lemma G.2** (Lemma C.6, Jin et al. 2020). Suppose Lemma F.2 holds, we have for all $(h,k) \in [H] \times [K]$,

$$V_h^k(s_h^k) - V_h^{\pi^k}(s_h^k) \leq V_{h+1}^k(s_{h+1}^k) - V_{h+1}^{\pi^k}(s_{h+1}^k)+$$
$$[PV_{h+1}^k - V_{h+1}^{\pi^k}](s_h^k, a_h^k) + 2\beta\|\phi(s_h^k, a_h^k)\|_{(\mathbf{U}_h^k)^{-1}},$$

where $V_{h+1}(\cdot) = 0$.

*Proof of Lemma F.4.* We first denote $\varepsilon_h^k$ as $[P(V_{h+1}^k - V_{h+1}^{\pi^k})](s_h^k, a_h^k)$, it's easy to verify that $\varepsilon_h^k$ is a zero-mean random variable conditioned on all randomness before $k$-th episode. By Lemma G.1, for all $(s,a,h,k) \in \mathcal{S} \times \mathcal{A} \times [H] \times [K]$, we have $Q_h^k(s,a) \geq Q_h^*(s,a) - 4H^2\zeta$. Also, by the definition of minimal sub-optimality gap in Definition 7.3, we have

$$\Delta_h(s_h^k, a_h^k) = V_h^*(s_h^k) - Q_h^*(s_h^k, a_h^k)$$
$$= Q_h^*(s_h^k, \pi_h^*(s_h^k)) - Q_h^*(s_h^k, a_h^k)$$
$$\leq Q_h^k(s_h^k, \pi_h^*(s_h^k)) + 4H^2\zeta - Q_h^*(s_h^k, a_h^k).$$

Since Algorithm 2 is taking the greedy policy $a_h^k = \operatorname{argmax}_a Q_h^k(s_h^k, a)$, the sub-optimality gap is bounded by

$$\Delta_h(s_h^k, a_h^k) \leq Q_h^k(s_h^k, a_h^k) - Q_h^*(s_h^k, a_h^k) + 4H^2\zeta \leq V_h^k(s_h^k) - V_h^\pi(s_h^k) + 4H^2\zeta, \quad (G.5)$$

since $V_h^{\pi^k}(s_h^k) \leq V_h^*(s_h^k)$. Following Lemma G.2 by telescoping we have

$$V_h^k(s_h^k, a_h^k) - V_h^{\pi^k}(s_h^k, a_h^k) = \sum_{h'=h+1}^{H} \varepsilon_{h'}^k + 2\beta \sum_{h'=h}^{H} \|\phi(s_{h'}^k, a_{h'}^k)\|_{(\mathbf{U}_{h'}^k)^{-1}}. \quad (G.6)$$

Plugging (G.6) back into (G.5) we will have the result as

$$\Delta_h(s_h^k, a_h^k) = 4H^2\zeta + \sum_{h'=h+1}^{H} \varepsilon_{h'}^k + 2\beta \sum_{h'=h}^{H} \|\phi(s_{h'}^k, a_{h'}^k)\|_{(\mathbf{U}_{h'}^k)^{-1}}.$$

Since it holds for all $k \in [K]$, we can take an additional summation over $k \in \mathcal{K} \subseteq [K]$ to get the claimed result. $\qquad\square$

