# OpenReview forum: "On the Interplay Between Misspecification and Sub-optimality Gap: From Linear Contextual Bandits to Linear MDPs"
_ICLR.cc/2023/Conference — Submitted to ICLR 2023_

### Official Review · Reviewer_Psq2 · 2022-10-23

**Confidence:** 4
**Correctness:** 3
**Technical Novelty And Significance:** 3
**Empirical Novelty And Significance:** 3
**Recommendation:** 5

**Clarity, Quality, Novelty And Reproducibility:**

The technical results of this paper are novel and clear. Although the main idea of this paper is easy to follow, some parts are not well-written.

**Strength And Weaknesses:**

Strength:

1. The idea of only using data with large uncertainty is well-motivated and novel.
2. The upper bound (Theorem 5.1) indicates that when the minimal sub-optimality gap is known, a constant regret bound is possible in the well-specified linear contextual bandits, which is an interesting byproduct.
3. This paper focuses on the gap-dependent bounds, which differs from the previous works that study gap-independent results.



Weakness:

1. For the proposed algorithm, its performance is only studied when the  misspecification level is small than $\Delta/(2\sqrt d \iota_1)$,  which may be a very small number. Although the following lower bound shows that  a worst-case linear regret is unavoidable when misspecification level is relatively large, it is still important to study the performance of DS-OFUL in general situations.
2. In the experiments, the authors only compare their algorithm with Lattimore et al. (2020). More methods from related works should be included.
3. The definition of $\beta$ in Theorem 5.1 is slightly different from that in Appendix C.



Minor comments:

1. Do you assume that the number of arms is finite in the lower bound part?
2. A few full stops are missing.
3. Sometimes $T$ rather than $K$ is used to denote the number of rounds, which is not consistent. Besides, in the literature of bandits, it is uncommon to use $K$ to denote the number of rounds.

**Summary Of The Paper:**

This paper studies the problem of regret minimization in misspecified linear contextual bandits. The proposed algorithm, named DS-OFUL, only learns the underlying parameter $\theta^*$ from arm pulling with large uncertainty, which makes it robust to misspecification. When the misspecification level is relatively small, a gap-dependent upper bound is proved, which does not depend on the number of time rounds. In addition, the authors provide a gap-dependent lower bound for the problem studied. It is shown that the misspecified linear contextual bandit model can be efficiently learnable only when the misspecification level is lower than a threshold $\tilde{\mathcal{O}}(\Delta / \sqrt{d})$. Finally, the authors extend their ideas into misspecified linear MDP and a gap-dependent upper bound is also proved.

**Summary Of The Review:**

This paper suggests an interplay between the misspecification level and the sub-optimality gap, which clearly tells when the misspecified linear contextual bandit model can be efficiently learnable. For the proposed algorithm, I believe a deeper understanding in general situations could be achieved. Overall, the contribution is technically sound and valuable though some issues need to be addressed (see the detailed comments above). Therefore, I vote for borderline.

---

> ### Author Response · Authors · 2022-11-14
> **Response to Reviewer Psq2**
>
> Thank you for your detailed comments!
>
> **Q1**: What happens when $\zeta > \Delta/\sqrt{d}$?
>
> **A1**: When $\zeta > \Delta / \sqrt{d}$, our algorithm can be reduced to OFUL by setting $\Gamma = 0$ to achieve gap-independent regret bound $O(d\sqrt{T} + \zeta \sqrt{d} T)$, which suggests the regret will grow as $\zeta$ increases.
>
> In addition, as our lower bound (Theorem 5.4) suggests, there exists a hard case where any algorithm will either 1) need to pull all arms or 2) suffer from a linear regret. Therefore, any algorithm leveraging the linear structure to avoid pulling all arms cannot have a sublinear cumulative regret depending on $\zeta$. Note that separating the case when \zeta < \Delta/\sqrt{d} (which leads to a constant regret) and \zeta > \Delta/\sqrt{d} (which gives a linear regret) is one of the major contributions in this paper, which sheds light on the interplay between the sub-optimality gap and the misspecification level.
>
> ***
>
> **Q2**: Add more experiments
>
> **A2**: Thank you for your suggestion. We have added a comparison with RLB in [Ghosh et al. (2017)](https://ojs.aaai.org/index.php/AAAI/article/view/11052) for misspecified bandits in Table 1 and Figure 1(a). The performance of RLB  is worse than OFUL and our algorithm since the misspecification level in our setting is relatively small.
>
> ***
>
> **Q3**: The definition of $\beta$.
>
> **A3**: This is indeed a typo. We have fixed it in the proof of Theorem 5.1
>
> ***
>
> **Q4**: T or K
>
> **A4**: We use K to unify the episode index in RL and time round in bandits. We have fixed the misuse of $T$ in some parts of our proof.

---

### Official Review · Reviewer_vLag · 2022-10-24

**Confidence:** 4
**Clarity, Quality, Novelty And Reproducibility:** The work can be improved if the above…
**Correctness:** 3
**Technical Novelty And Significance:** 3
**Empirical Novelty And Significance:** 2
**Recommendation:** 5

**Strength And Weaknesses:**

Strength:
1. The work is well-organized and easy to follow.
1. It is novel to discard data that brings little uncertainty to the model, which improves the robustness of the algorithm in the misspecified setting.
1. It provides both a high-probability and an expected bound on the regret in the linear bandit setting.

Weaknesses:
1. The experiments in linear bandits are not so convincing. Firstly, why do the author(s) apply such a grid search for the parameters? The data shown in the work does not represent a "uniform search". Secondly, what is the theoretical explanation of the fact that the algorithm with $\Gamma=0.08$ performs the best?
1. As the algorithm is sensitive to the value of $\Gamma$, how should the value of this parameter be selected?
1. Is it possible to provide some numerical results for the linear MDPs?


**Summary Of The Paper:**

This work focuses on the misspecification setting for both linear contextual bandits and linear MDPs.
For the linear bandits, it proposes the DS-OFUL algorithm. It derives an upper bound accordingly and also a lower bound.
For the linear MDPs, it devises the DS-LSVI with a similar design and also provides un upper bound for it.
It also provides numerical results in the linear bandit setting.

**Summary Of The Review:**

This work considers the misspecification setting in the bandits. It proposes and analyzes a novel algorithm: DS-OFUL. It also extends the design to the linear MDPs and analyzes the DS-LSVI algorithm. The work makes some contribution, which would be presented cleared if the weaknesses above can be improved.


===============


After rebuttal:
Thanks for the rebuttal. I actually wonder if the optimal hyper parameters can be found in a quick grid search in a more difficult instance. Besides, $\Delta$ may not always be prior knowledge.

I would like to keep the score.

---

> ### Author Response · Authors · 2022-11-14
> **Response to Reviewer vLag**
>
> Thank you for your helpful comments. We address your questions and concerns as follows.
>
> **Q1**: Why the grid search is not uniform? The theoretical explanation of Γ=0.08 performs the best
>
> **A1**: In our experiments, the hyperparameters are tuned by the uniform search on a logarithmic grid (e.g.,  $\lambda = 10^0, 10^{0.5}, 10^{1}$). Searching the hyperparameters in a logarithmic grid is more efficient than searching in a linear grid. Per your suggestion, on the synthetic dataset, we set $\Gamma = \Delta / \sqrt{d} = 0.2 / \sqrt{16} \approx 0.05$ and the result is similar to the one with $\Gamma = 0.08$.
>
> We do not have a theoretical calculation showing that $\Gamma = 0.08$ is the best choice since $\Gamma = O(\Delta / \sqrt{d})$ hides the constants and the log factors. Nevertheless, as the experiment results suggest, it’s easy to find a good $\Gamma$ using grid search.
>
> ***
>
> **Q2**: How to choose the threshold parameter in the algorithm.
>
> **A2**: If the sub-optimal gap is known due to the nature of the problem or a predefined tolerance of the sub-optimal gap (i.e., we tolerate the sub-optimality below a certain threshold), then our experiments show that $\Gamma = \Delta / \sqrt{d}$ is a good choice. Otherwise, we can tune $\Gamma$ using grid search since a larger $\Gamma$ will directly lead to a linear regret.
>
> ***
>
> **Q3**: Add experiments on Linear MDPs
>
> **A3**: We have added the experiments on Linear MDPs in Appendix B.4. The experimental results show that adding a properly chosen $\Gamma$ will reduce the cumulative regret, and the best choice of $\Gamma$ is consistent with our theoretical analysis.

---

### Official Review · Reviewer_1suH · 2022-10-27

**Confidence:** 3
**Clarity, Quality, Novelty And Reproducibility:** 1)Author motivates the subject from a…
**Correctness:** 4
**Technical Novelty And Significance:** 3
**Empirical Novelty And Significance:** 3
**Recommendation:** 6

**Strength And Weaknesses:**

1)Author motivates the subject from a theoretical and practical point of view. 2)In the introduction, the author reviewed the previous works carefully. 3)the author successfully addressed the issue and recommended the algorithms. 4)The authors successfully provide simulation results for their algorithm. 5)The proof is mathematically correct.

**Summary Of The Paper:**

This paper looks at a linear contextual bandit and linear MDP in a misspecified setting. They provide an algorithm called DS-OFU for these problems. They showed that an $O(d^2/\Delta)$ regret is achievable if the $\zeta$ is small.


**Summary Of The Review:**

this paper is well-written and addresses an interesting problem.

---

> ### Author Response · Authors · 2022-11-14
> **Response to Reviewer 1suH**
>
> Thank you for your positive feedback.

---

### Official Review · Reviewer_2Y3y · 2022-10-29

**Confidence:** 3
**Correctness:** 3
**Technical Novelty And Significance:** 3
**Empirical Novelty And Significance:** 3
**Recommendation:** 6

**Clarity, Quality, Novelty And Reproducibility:**

The paper is quite clearly written. The DS technique is novel and the corresponding algorithms it yields. The corresponding regret lower and upper bounds are also interesting. The algorithms and proofs are described with enough details.

**Strength And Weaknesses:**

Strength:
The problem and results as mentioned in the summary are interesting and relevant to the existing literature in this topic.

Weakness/Questions:
1. The claim that the regret bound improves the logarithmic factor log (horizon) than existing linear contextual bandits results is misleading. For \delta=1/K, which is a common and justified choice, we observe that this term reappears. Removing this claim misleads more than focusing on other interesting contributions in this paper.

2. What happens to the regret upper bound of DS-OFUL when \zeta > \Delta/\sqrt{d}? How does it grow with \zeta or is it still robust up to certain limit?

3. What is a good way to tune \Gamma than trying different values as done in experiments? Does the analysis point us to any such choice?

4. In experiments, we miss two things: a comparison with other misspecified bandit algos (such as Ghosh et al., 2017) and an experiment on linear MDPs. This leads to two questions: How is this algorithm better/worse than misspecified bandit algos? What is the challenge in applying DS-LSVI to lin MDPs and what benefit do we get in that case?

**Summary Of The Paper:**

The paper studies linear contextual bandits and linear MDPs with misspecifications. It proposes a new thresholding scheme for data selection, which can be used with OFUL and LSVI seamlessly. For linear contextual bandit, DS-OFUL yields regret upper bounds of same order as that of the well-specified setting. The paper also provides a lower bound on regret for higher levels of misspecifications. For lin MDPs, it proposes DS-LSVI, which achieves logarithmic problem dependent regret.

**Summary Of The Review:**

The paper theoretically studies misspecification in LCBs and LMDPs. It proposes a data selection refinement with OFUL and LSVI, that only considers data with higher uncertainty. The corresponding regret analysis conforming the bound on misspecification level, i.e. O(\Delta/\sqrt{d}), is an interesting result. The paper still lacks some portions as pointed out in the weakness. Addressing them will strengthen understanding of the contributions.

*****
After rebuttal: The authors have improved the experimental section with new comparisons demonstrating effectiveness of DS-OFUL and DS-LSVI. But the idea to tune $\Gamma = \Delta/ \sqrt{d}$ seems unrealistic, as $\Delta$ is not a prior knowledge in real applications. Additionally, the algorithm seems to be quite sensitive to $\Gamma$ and can behave similar to (Lattimore et al., 2020) if not tuned. Thus, I shall increase my score due to the response but cannot argue for an acceptance.

---

> ### Author Response · Authors · 2022-11-14
> **Response to Reviewer 2Y3y**
>
> Thank you for your constructive comments. We address your questions as follows.
>
> **Q1**: the claim on the improvement of logarithmic factor.
>
> **A1**: We want to clarify that our constant regret bound, even depending on $log(1/\delta)$, is highly non-trivial to obtain. Recall that in [Abbasi-Yadkori et al. (2011)](https://papers.nips.cc/paper/2011/hash/e1d5be1c7f2f456670de3d53c7b54f4a-Abstract.html), their regret bound is of $d^2\log(K)\log(1 / \delta)$. Therefore, even when $\delta$ is a constant, e.g., $\delta = 1 \times 10^{-3}$, their regret bound still depends on $\log(K)$. In sharp contrast, our regret bound is indeed a constant regret when $\delta$ is a constant. Therefore, the logarithmic improvement in $K$ is indeed a nontrivial contribution. Similar logarithmic improvement in $K$ has also been achieved by [Papini et al. (2021)](https://proceedings.mlr.press/v139/papini21a.html), but under an additional assumption on the underlying context distribution. Our improvement does not rely on such an additional assumption but is due to the algorithmic design of data selection.
>
> ***
>
> **Q2**: What happens when $\zeta > \Delta/\sqrt{d}$?
>
> **A2**: When $\zeta > \Delta / \sqrt{d}$, our algorithm can be reduced to OFUL by setting $\Gamma = 0$ to achieve gap-independent regret bound $O(d\sqrt{T} + \zeta \sqrt{d} T)$, which suggests the regret will grow as $\zeta$ increases.
>
> In addition, as our lower bound (Theorem 5.4) suggests, there exists a hard case where any algorithm will either 1) need to pull all arms or 2) suffer from a linear regret. Therefore, any algorithm leveraging the linear structure to avoid pulling all arms cannot have a sublinear cumulative regret depending on $\zeta$. Note that separating the case when $\zeta < \Delta/\sqrt{d}$ (which leads to a constant regret) and $\zeta > \Delta/\sqrt{d}$ (which gives a linear regret) is one of the major contributions in this paper, which sheds light on the interplay between the sub-optimality gap and the misspecification level.
>
> ***
>
> **Q3**: How to choose the threshold parameter in the algorithm.
>
> **A3**: If the sub-optimal gap is known due to the nature of the problem or a predefined tolerance of the sub-optimal gap (i.e., we tolerate the sub-optimality below a certain threshold), then our experiments show that $\Gamma = \Delta / \sqrt{d}$ is a good choice. Otherwise, we can tune $\Gamma$ using grid search since a larger $\Gamma$ will directly lead to a linear regret.
>
> ***
>
> **Q4**: Add more experiments
>
> **A4**: Thank you for your suggestion. We have added a comparison with RLB in [Ghosh et al. (2017)](https://ojs.aaai.org/index.php/AAAI/article/view/11052) for misspecified bandits in Table 1 and Figure 1(a). The performance of RLB is worse than OFUL and our algorithm. We have also added the experiments on Linear MDPs in Appendix B.4. The experimental results show that adding a properly chosen $\Gamma$ will reduce the cumulative regret, and the best choice of $\Gamma$ is consistent with our theoretical analysis.

---

### Official Review · Reviewer_Btqn · 2022-10-31

**Confidence:** 3
**Correctness:** 4
**Technical Novelty And Significance:** 3
**Empirical Novelty And Significance:** 3
**Recommendation:** 6

**Clarity, Quality, Novelty And Reproducibility:**

The main ideas are clearly laid out with proof sketches throughout the paper . The data selection is novel and avoids the linear regret with the caveat of a known delta. Also, the main algorithms are shown in the paper (Algorithm 1, 2) and can be easily reproduced.

**Strength And Weaknesses:**

Pros:
   (A) Novel algorithm for data selection such that it learns from the uncertain data points to obtain a O(d^2/Delta) gap-dependent regret bound when the misspecification is small. Similarly, they show a matching lower bound for learnability for the corresponding low misspecification level and not efficiently learnable otherwise. Similar results are shown for the misspecified linear MDP settings to achieve a logarithmic regret bound.
   (B) They show extensive experiments on both synthetic and real datasets. On the synthetic dataset, they show that their regret for their proposed algorithm DS-OFUL is better than the one from Lattimore et al. Additional experiments showcasing the performance benefits are shown in supplementary.

Cons:

   (i) It is not clear how the threshold parameter is chosen in the algorithms? Does it not depend on the data and needs to be adaptively estimated and refined?


**Summary Of The Paper:**

The paper studies the linear contextual bandits in the misspecified setting where the reward function can be approximated by a linear function class up to a misspeci
fication level. In the case of the misspecification being smaller than the gap between the best and second best arm (delta) over the square root of the dimension, then they show an efficiently learning algorithm with a constant regret bound (better by a log(K) factor). Experiments are shown on both simulated and real-world settings comparing their proposed approach with a previous algorithm.

**Summary Of The Review:**

Overall, an interesting data selection idea applied to both the linear contextual bandit and linear MDP in the misspecified settings with good regret guarantees. Some lingering questions remain and hope to discuss with the authors.

---

> ### Author Response · Authors · 2022-11-14
> **Response to Reviewer Btqn**
>
> Thank you for your positive comments! We address your question as follows.
>
> **Q**: How to choose the threshold parameter in the algorithm.
>
> **A**: If the sub-optimal gap is known due to the nature of the problem or a predefined tolerance of the sub-optimal gap (i.e., we tolerate the sub-optimality below a certain threshold), then our experiments show that $\Gamma = \Delta / \sqrt{d}$ is a good choice. Otherwise, we can tune $\Gamma$ using grid search since a larger $\Gamma$ will directly lead to a linear regret.

---

### Author Response · Authors · 2022-11-14
**To all reviewers**

Thank you for all your valuable comments. We have updated our manuscript and here is a brief summary of the revisions. The major revisions are marked in red in the paper.

1. We have added an experiment on Linear MDPs, which shows that our algorithm DS-LSVI could outperform the LSVI-UCB by choosing a good $\Gamma$. The choice of $\Gamma$ is well aligned with our theoretical analysis (Appendix B.4)
2. We have added a comparison with the RLB algorithm in Ghosh et al. (2017) for misspecified linear bandits in Figure 1(a) and Table 1. The result shows that RLB performs worse than OFUL and our algorithm since the misspecification level in our setting is relatively small (See Section 8.1).
3. We have fixed the typos in the paper.

---

### Decision · Program_Chairs · 2023-01-20

**Decision:**

Reject

**Justification For Why Not Higher Score:**

Algorithm not parameter free. See above.

**Justification For Why Not Lower Score:**

Can't go lower.

**Metareview: Summary, Strengths And Weaknesses:**

The authors consider a misspecified linear bandit and linear MDP problem in which the reward can be approximated by a linear function up to an unknown misspecification parameter $\zeta$. The authors then considered the interplay between the $\zeta$ and the suboptimality gap $\Delta$ on the regret. Upper and lower bounds on the regret are provided. The paper is generally well written with a well-motivated problem statement and sound analyses.

This is a borderline paper that warranted a discussion among the reviewers, which I conducted. Several of the reviewers had recurring concerns about the selection of the threshold $\Gamma$ in the algorithm and the fact that the authors assume that the suboptimality gap $\Delta$ is known. The authors tune $\Gamma = d/\sqrt{\Delta}$ by grid search, but $\Delta$ is not known in practice. I believe that these are deficiencies in the paper that need to be addressed before it can be published. The necessity of $\zeta$ being smaller than $\Delta/\sqrt{d}$ should be emphasized.

**Summary Of Ac-Reviewer Meeting:**

We had an email discussion and the reviewers were unanimous in their assessment that the algorithm is not parameter free, which is a major deficiency.